# Sampling-Efficient Test-Time Scaling: Self-Estimating the Best-of-$N$ Sampling in Early Decoding

**Yiming Wang**[α]  **Pei Zhang**[β]  **Siyuan Huang**[α]  **Baosong Yang**[β]
**Zhuosheng Zhang**[α]  **Fei Huang**[β]  **Rui Wang**[α, ✉]

[α]School of Computer Science, Shanghai Jiao Tong University
[β]Tongyi Lab, Alibaba Group
✉: Corresponding Author
Email: [α]{yiming.wang, wangrui12}@sjtu.edu.cn

## Abstract

Test-time scaling enhances large language model performance by allocating additional compute resources during inference. Best-of-$N$ (BoN) sampling serves as a common sampling-based scaling technique, broadening the search space in parallel to find better solutions from the model distribution. However, its cost–performance trade-off is still underexplored. Two main challenges limit the efficiency of BoN sampling: (1) Generating $N$ full samples consumes substantial GPU memory, reducing inference capacity under limited resources. (2) Reward models add extra memory and latency overhead, and training strong reward models introduces potential training data costs. Although some studies have explored efficiency improvements, none have addressed both challenges at once.

To address this gap, we propose **Self-Truncation Best-of-$N$ (ST-BoN)**, a decoding method that avoids fully generating all $N$ samples and eliminates the need for reward models. It leverages early sampling consistency in the model's internal states to identify the most promising path and truncate suboptimal ones. In terms of cost, ST-BoN reduces dynamic GPU memory usage by over 80% and inference latency by 50%. In terms of cost–performance trade-off, ST-BoN achieves the same performance as Full-BoN while saving computational cost by 70%–80%, and under the same cost, it can improve accuracy by 3–4 points.

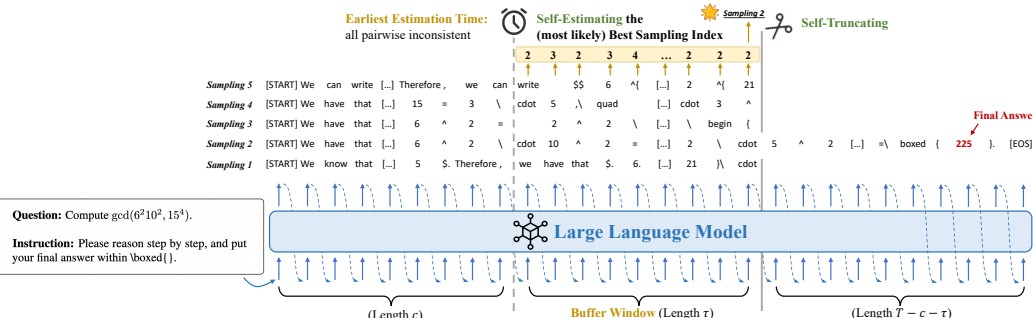

Figure 1: The pipeline and case of our **Self-Truncation Best-of-$N$ (ST-BoN) Decoding**. From the start: **(Step 1)** We first let the LLM generate $N$ samplings autoregressively until the earliest estimation time $c$; **(Step 2)** Then each sampling continues to generate $\tau$ steps beyond time $c$. During this, the LLM performs self-estimation to determine the most promising sample among the $N$ sampling, identifying the best sample through cumulative $\tau + 1$ self-estimations. **(Step 3)** Finally, we truncate the remaining $N - 1$ samples and proceed with generating only the best-estimated sample up to [EOS]. Parameter details of the case in this figure: $N = 5$, $c = \tau = 20$, and $T = 359$.

# 1 Introduction

Large Language Models (LLMs) possess strong generative and reasoning capabilities after extensive training on large-scale corpora [7, 1, 10], and the excellent solution to a problem is typically embedded within the model distribution [26, 9, 48, 53, 30]. However, autoregressive decoding focuses on locally optimal solutions, overlooking more promising thinking paths. Hence, the test-time scaling technique [40] is proposed to help LLMs expand their thinking space by adding compute during inference.

Best-of-$N$ (BoN) [42, 28, 40] serves as a widely used sampling-based scaling paradigm. By sampling $N$ responses from the LLM and selecting the best one with a re-ranking strategy at inference time, BoN fully leverages the potential ability within the model distribution. The core of BoN sampling lies in scoring and re-ranking multiple candidates. A well-known method is self-consistency [47], which selects the most frequent answer as the final solution. Furthermore, with a trained reward model (RM) [45, 46, 35], BoN sampling can handle richer tasks by scoring candidates and selecting the one with the highest score, aiming to increase test-time computational costs regardless of expense to enhance performance [40].

Beyond performance, the issue of cost-performance trade-offs in BoN sampling has been less explored. Under existing paradigms, BoN sampling faces two key challenges that hinder efficient deployment: (i) *Full Generation Overhead*: Traditional BoN requires fully generating all $N$ samples, referred to as **Full-BoN** in this paper. Although GPU parallelization can partially mitigate time latency, *additional memory overhead is unavoidable*, especially for complex reasoning with long generated sequences. (ii) *Limitations of Reward Models*: RMs are effective, but *occupy additional memory and time costs*. Meanwhile, training a strong RM is costly due to the scarcity of high-quality feedback data, and their domain-specific nature restricts generalizability across tasks (*e.g.*, from math to open-ended QA).

Some efforts have been made to improve the efficiency of BoN sampling by using RMs to score and discard sampling prefixes [41], or to branch after pruning [25, 36], thereby reducing some memory overhead of full generation. However, RM inference still incurs non-negligible latency, and sharing GPU memory with generators further restricts caching capacity. Moreover, the inherent limitations of RMs make these methods less suitable for objective tasks like reasoning (see Appendix B). Therefore, *no existing method overcomes both challenges at once to achieve deeper efficiency optimizations*.

To fill this gap, we introduce a novel decoding technique called **S**elf-**T**runcation **B**est-**o**f-$N$ (**ST-BoN**). Inspired by the unsupervised advantage of consistency strategies [47], we first theoretically demonstrate the feasibility of foreshadowing the final consistency using sampling consistency in early decoding. Then, we design an internal consistency measure that uses latent space information [49] to identify and preserve the most promising sample among the $N$ samples in early decoding. This framework allows us to effectively self-estimate and truncate suboptimal samplings in early decoding without relying on reward model intervention, thereby saving memory and speeding up generation.

The detailed pipeline of ST-BoN is illustrated in Figure 1, it makes the following contributions[1]:

- *Cost:* Compared to Full-BoN, ST-BoN truncates suboptimal samples in early decoding, thereby freeing up memory and **reducing the dynamic GPU memory load by over 80%**. This early truncation also accelerates later generations, **reducing inference latency by up to 50%**. Notably, these efficiency gains are achieved without accounting for the overhead of extra reward models.

- *Cost-Performance Trade-offs:* We conduct extensive experiments to demonstrate that ST-BoN strikes a strong balance between cost and performance. ***When reaching the performance of Full-BoN at the certain $N$ values*, ST-BoN can save computational cost by 70% to 80%**. Also, ***when consuming similar computational costs*, ST-BoN improves accuracy by 3 to 4 points**. In addition, ST-BoN is applicable across a wide range of domains. These results show that ST-BoN can provide a flexible solution to balance cost and performance when faced with limited resources.

# 2 Preliminary

**Autoregressive Decoding in Language Models.** Let $p_\theta$ represent a language model. For a given prompt and question, $p_\theta$ will generate a token sequence $Y_{1:T} = y_1 y_2 ... y_T$ autoregressively, where $T$

---

[1]Code and data are available in `https://github.com/Alsace08/ST-BoN`.

is the generation length and $y_T = $ [EOS]. Each token $y_t$ is sampled as follows:

$$y_t \sim p_\theta(\cdot | \text{prompt, question}, y_{\prec t}) \in \mathbb{R}^{|\mathcal{V}|}, \tag{1}$$

where $\mathcal{V}$ is the vocabulary. In greedy decoding, the model selects the token with the highest probability of $p_\theta$ to generate $y_t$. In contrast, sampling decoding typically uses multinomial sampling, such as top-$k$ [13] or top-$p$ [20], to let models sample $y_t$ from $p_\theta$ after truncation to generate diverse sequences.

**Best-of-$N$ Sampling.** As the name suggests, Best-of-$N$ sampling involves sampling $N$ sequences to find the best one. Let $Y^1, Y^2, \ldots, Y^N$ be $N$ independent sampling sequences, then each sample $Y^i$ is assigned a scalar score $S(Y^i)$, and the best sample index is identified as $\arg\max_i \{S(Y^i)\}_{i=1}^N$.

The simplest scoring approach is to use an externally trained reward model. If avoiding the intervention of reward models, the **consistency strategy** is the most common unsupervised alternative, introduced by self-consistency [47], which found a positive correlation between response consistency and accuracy; that is, when multiple reasoning paths lead to the same answer, confidence in its correctness increases. In self-consistency, the exact answer $A^i$ is extracted from $Y^i$, and the consistency score $S(Y^i)$ is computed based on the frequency with which $A^i$ co-occurs with other answers:

$$S(Y^i) = \frac{\sum_{j=1, j \neq i}^N \mathbb{I}(A^i = A^j)}{N - 1}, \tag{2}$$

where $\mathbb{I}(\cdot)$ is the indicator function. More generally, "consistency" can be any continuous measure. For example, [21] suggested using semantic similarity to assess sample consistency in open-ended tasks, calculated through the distance between sentence embeddings.

## 3 ST-BoN: Self-Truncation Best-of-$N$ Decoding

### 3.1 Theoretical Support: Early Consistency Foreshadows Final Consistency

We aim to overcome the efficiency bottlenecks of both full generation and reward models. Without a reward model, consistency strategies offer a promising unsupervised approach to select optimal samples. However, existing consistency methods require full generation. Naturally, we hope that early consistency can foreshadow the final consistency: *a sample closer to others in early decoding is more likely to reach the correct answer when the decoding ends*. If this holds, such consistency propagation can be utilized to estimate the most promising sampling in early decoding.

Intuitively, considering the cumulative effect of autoregressive generation, sampling sequences that already exhibit large differences in early decoding is less likely to converge closely by the end. Theoretically, this can be modeled as probabilistic monotonicity: *as early consistency increases, the probability lower bound for final consistency surpassing a given constant also increases*.

Consistency can be mathematically converted into a distance measure, where higher consistency implies a smaller distance. Let $\mathcal{D}(\cdot, \cdot) : Y \times Y \to \mathbb{R}^+$ be an arbitrary metric function defined over a metric space, used to quantify the distance between two partial sequences $Y_{1:t}^i$ and $Y_{1:t}^j$ at time $t$. We demonstrate that early consistency can probabilistically enhance the likelihood of final consistency. Specifically, let $Y^1$ denote the primary sampling. At an early decoding time $t$ ($t < T$), if the distance between the partial sequence $Y_{1:t}^1$ and the other $N - 1$ partial sequences $\{Y_{1:t}^i\}_{2 \leq i \leq N}$ becomes smaller under a given metric $\mathcal{D}$, the lower bound of the probability that the distance between the final full sequence $Y_{1:T}^1$ and the other $N - 1$ full sequences $\{Y_{1:T}^i\}_{2 \leq i \leq N}$ is less than a certain value will increase, meaning it is more likely to meet a specific consistency requirement.

**Theorem 1.** *Let $d_t^i = \mathcal{D}(Y_{1:t}^1, Y_{1:t}^i)$ denote the distance between sampling sequence $Y_{1:t}^1$ and the $i$-th sampling sequence $Y_{1:t}^i$ at decoding time $t$. For a given constant $\epsilon$, there exist a constant $\Gamma$ such that:*

$$\Pr[S_T \leq \epsilon \mid S_t] \geq 1 - \frac{\Gamma^{T-t}}{\epsilon} S_t,$$

*where $1 \leq t < T$, $S_T = \sum_{i=2}^N d_T^i$ and $S_t = \sum_{i=2}^N d_t^i$.*

*Proof Sketch.* We assume (i) local Lipschitz continuity, *i.e.*, for any two prefixes, the next-token distributions differ in TV distance by at most $L$ times their sequence distance, and (ii) bounded

increments, so appending a single token can increase the distance by at most $M$. Under maximal coupling, two continuations diverge at step $t + 1$ with probability at most $Ld_t^i$. Thus the expected distance evolves as $\mathbb{E}[d_{t+1}^i \mid d_t^i] \leq d_t^i + M(Ld_t^i) = (1 + LM)\, d_t^i$. Summing over $i$ yields $\mathbb{E}[S_{t+1}] \leq \Gamma S_t$ with $\Gamma = 1 + LM$. Iterating this recursion shows that deviations grow at most exponentially: $\mathbb{E}[S_T] \leq \Gamma^{T-t} S_t$. Finally, applying Markov's inequality converts the expectation bound into a high-probability guarantee: $\Pr[S_T \geq \epsilon \mid S_t] \leq \Gamma^{T-t} S_t / \epsilon$, implying the claim in Theorem 1. $\square$

The detailed proof is shown in Appendix A. As $S_t$ increases, the probability lower bound of $S_T \leq \epsilon$ also increases. This confirms the potential for early consistency that foreshadows the final consistency.

## 3.2 Method: Early Internal Consistency Estimates the Most Promising Sampling

We have proven the high-probability foreshadow from early consistency to final consistency, providing theoretical feasibility for utilizing sampling consistency in early decoding to estimate the most promising path. Upon this, we are ready to present the main ST-BoN algorithm, which involves defining an effective measure of sampling consistency by designing a suitable metric function $\mathcal{D}$.

**When Does Self-Estimation Start?**  Before defining the measure, we first identify the earliest self-estimation time $t$ for early consistency. We aim to achieve ultimate efficiency improvement, so the earliest time can occur when *all samplings become pairwise inconsistent*. Let this special time be $c$, it satisfies the following condition:

$$\sum_{i,j,i \neq j} \mathbb{I}\left(Y_{1:c}^i = Y_{1:c}^j\right) = 0 \quad \& \quad \forall t < c, \ \sum_{i,j,i \neq j} \mathbb{I}\left(Y_{1:t}^i = Y_{1:t}^j\right) > 0. \tag{3}$$

In practical inference, we enable GPU parallelization and perform pairwise sequence-equality checks at each time. If any sample reaches [EOS] during this process, it is terminated immediately, and the current time step is recorded as $c$. In practice, this situation is very rare, as the condition in Eq.3 is typically satisfied early on, as shown in Figure 3.

**Main Algorithm: How is Self-Estimation Done?**  Now we start to define an effective measure of sampling consistency. At such an early decoding stage, semantic information is limited, and textual differences between samplings are minimal, making it hard to distinguish them solely based on output text. Therefore, we consider utilizing the more informative hidden states of LLMs to represent early sampling sequences through the model's internal information.

Chain-of-Embedding (CoE) offers a promising idea, which is the representation technique in the latent space [49] that links all hidden states from input to output, capturing the *latent thinking path* from reading to writing. It extracts hidden states across layers to form a progressive chain. Consider a model with $L$ layers and an output length $T$, then the hidden state at layer $l$ ($0 \leq l \leq L$) and position $t$ ($1 \leq t \leq T$) is represented as $z_t^l$. The sentence embedding for layer $l$ is calculated as $h_l^T = \frac{1}{T}\sum_{t=1}^T z_t^l$ [38, 50]. Thus, CoE is defined as a progressive chain of sentence embeddings: $\boldsymbol{H}_{1:T} := h_0^T \to h_1^T \to \cdots \to h_L^T$. The CoE feature $\mathcal{F}(\boldsymbol{H})$ is quantified as:

$$\mathcal{F}(\boldsymbol{H}_{1:T}) = \frac{1}{L} \cdot \sum_{l=0}^{L-1} \left( \frac{M(h_l, h_{l+1})}{M(h_0, h_L)} - \frac{A(h_l, h_{l+1})}{A(h_0, h_L)} \right), \tag{4}$$

where $M(h_i, h_j) = ||h_i - h_j||_2$ and $A(h_i, h_j) = \arccos\left(h_i^\top \cdot h_j / (||h_i||_2 \cdot ||h_j||_2)\right)$. A larger $\mathcal{F}(\boldsymbol{H}_{1:T})$ indicates a greater curvature in the latent thinking path when generating $Y_{1:T}$.

Using the CoE measure, we define the distance $\mathcal{D}(Y_{1:c}^i, Y_{1:c}^j)$ between two partial sequences $Y_{1:c}^i$ and $Y_{1:c}^j$ at time $c$ as the squared difference of their CoE features. The consistency score $S(Y_{1:c}^i)$ is then calculated as the average distance between the $Y_{1:c}^i$ and the other $N - 1$ samples. Specifically:

$$\mathcal{D}(Y_{1:c}^i, Y_{1:c}^j) = \left( \mathcal{F}(\boldsymbol{H}_{1:c}^i) - \mathcal{F}(\boldsymbol{H}_{1:c}^j) \right)^2, \quad S(Y_{1:c}^i) = \frac{1}{N-1} \sum_{j=1, j \neq i}^N \mathcal{D}(Y_{1:c}^i, Y_{1:c}^j). \tag{5}$$

A smaller $S(Y_{1:c}^i)$ indicates higher consistency between the $i$-th sample and the other samples. After computing scores for all $N$ samples, we rerank them and choose the sample with the lowest score: $i_c = \arg\min_i \{S(Y_{1:c}^i)\}_{i=1}^N$, making the $i_c$-th sample the optimal estimate at time $c$.

**How Long Does Self-Estimation Last?** Performing self-estimation at a single moment can introduce randomness, since pairwise sequence differences only begin to emerge at time $c$, and these differences may not be substantial. To migrate this, we define a **Buffer Window**: *the LLM continues to generate for an additional $\tau$ steps beyond time $c$, where $\tau$ is the window length.* We set $\tau \propto c$, *i.e.*, $\tau = mc$, where $m$ is a proportional constant. The intuition is that a smaller $c$ indicates an early divergence between samplings, which reflects greater randomness, so a smaller window may suffice to capture later differences. In contrast, a larger $c$ suggests lower randomness, which requires a larger window to capture sufficient later differences. At each time $c'$ within the buffer window, we obtain the optimal estimation $i_{c'} = \arg\min_i \{s(Y^i_{1:c'})\}^N_{i=1}$ using Eq.5. This produces $\tau + 1$ optimal estimation $\{i_{c'}\}^{c+\tau}_{c'=c}$, and the final optimal estimation $i_{\text{final}}$ is determined by selecting the most frequent one:

$$\text{Count}(i) = \sum_{t=c}^{c+\tau} \mathbb{I}(i_t = i) \ \ (1 \le i \le N), \qquad i_{\text{final}} = \arg\max_i \{\text{Count}(i)\}^N_{i=1}. \qquad (6)$$

# 4 Cost Analysis: Towards Efficient Best-of-$N$ Sampling

Compared to Full-BoN, ST-BoN reduces significant computational cost during inference, which is reflected in two aspects: **Space** and **Time**. The space is related to GPU memory overhead, and the time is reflected in inference latency. We will analyze the two parts between ST-BoN and Full-BoN.

## 4.1 Inference Overhead: GPU Memory

During model inference, GPU memory overhead is mainly impacted by the *model weights* and the *KV cache*. While model weight loading is necessary, the KV cache is the main memory bottleneck. Autoregressive parallel sampling can quickly cause Out-of-Memory (OOM) issues due to KV cache accumulation, with OOM events tied to peak memory overhead. Therefore, we assess ST-BoN's memory optimization by comparing its reduction in peak memory overhead to that of Full-BoN.

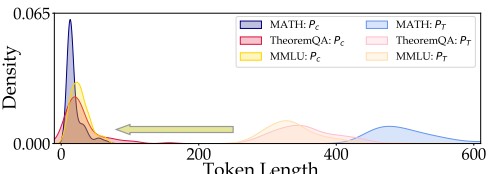

Figure 2: The distributions $P_c$ and $P_T$ of the earliest estimation time $c$ and the full generation length $T$ across different datasets.

Assuming that $N$ samples are generated in parallel on the GPU (*i.e.*, the batch size is $N$), the KV cache usage is linearly related to the sampling size $N$ and the generation length $T$. In the dataset $\mathcal{D}$, for inputs $X \sim \mathcal{D}$ and outputs $Y \sim p_\theta(\cdot|X)$, we let $P_T$ be the length distribution of full generation, and $P_c$ be the distribution of the earliest estimation time $c$. When excluding basic model weight memory: (i) For Full-BoN without reward models, peak memory occurs at time $T$ with batch size $N$. Note that if a reward model is added, the basic memory overhead of Full-BoN will increase, further reducing the available inference space. (ii) For ST-BoN, the peak memory can occur at time $c$ with batch size $N$, or at time $T$ with batch size 1. We set $m = 1$ in our main experiments, so that $\tau = c$, and the memory reduction rate $R_\mathcal{D}$ on dataset $\mathcal{D}$ can be approximated as:

$$R_\mathcal{D} = 1 - \frac{\max\left\{N \cdot (\mathbb{E}_{c \sim P_c}[c] + \tau), \mathbb{E}_{T \sim P_T}[T]\right\}}{N \cdot \mathbb{E}_{T \sim P_T}[T]} = 1 - \max\left\{2 \cdot \frac{\mathbb{E}_{c \sim P_c}[c]}{\mathbb{E}_{T \sim P_T}[T]}, \frac{1}{N}\right\}. \qquad (7)$$

We conduct statistical experiments on the distributions $P_c$ and $P_T$ using the Llama3-8B-Instruct model across three datasets: MATH [18], TheoremQA [8], and MMLU [17]. Figure 2 visualizes the respective distributions, showing a clear shift to the left of $P_c$ compared to $P_T$. This indicates that the earliest estimation time $c$ appears much earlier than the completion of sampling.

Furthermore, we calculate $\mathbb{E}_{c \sim P_c}[c]$ and $\mathbb{E}_{T \sim P_T}[T]$, set various sampling sizes $N$, and use Eq.7 to find $R_\mathcal{D}$. Figure 3 illustrates that $R_\mathcal{D}$ **easily surpasses 80%** for $N \geq 5$. According to Eq.7, the upper limit of $R_\mathcal{D}$ is

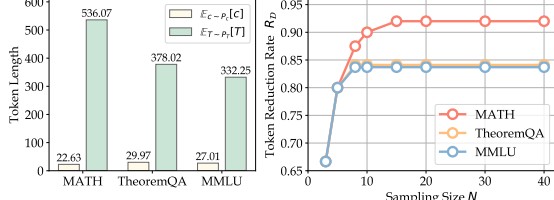

Figure 3: **(Left)** The mathematical expectations $\mathbb{E}_{c \sim P_c}[c]$ and $\mathbb{E}_{T \sim P_T}[T]$ of the earliest estimation time $c$ and the full generation length $T$ across different datasets; **(Right)** The computed memory reduction rate $R_\mathcal{D}$ under different sampling sizes $N$.

$1 - 2\mathbb{E}_{c \sim P_c}[c]/\mathbb{E}_{T \sim P_T}[T]$, and increases with the sampling size $N$ until it reaches this limit. In addition, significant variations in $\mathbb{E}_{T \sim P_T}[T]$ compared to minimal variations in $\mathbb{E}_{c \sim P_c}[c]$ make domain optimization primarily dependent on $\mathbb{E}_{T \sim P_T}[T]$. For example, the MATH dataset has a longer full generation, leading to a higher optimization upper limit than other domains. **In summary, $R_\mathcal{D}$ increases with the sampling size $N$ and the complexity of the task**.

## 4.2 Inference Latency: Wall-Clock Time

Inference latency is another critical aspect that affects computational cost. In Full-BoN, while $N$-sampling can run in parallel on the GPU, it still introduces more delay compared to single-sample inference. Our ST-BoN method leverages early self-truncation, enabling LLMs to resume single-sample generation after time $c + \tau$. However, sequence equality checks and self-estimation within the buffer window require serial vector processing, potentially adding latency. To address this, we will evaluate inference latency using wall-clock time as the measure.

We perform statistical experiments using the Llama3-8B-Instruct model on the MATH and TheoremQA datasets. For Full-BoN, we assess scenarios with and without a reward model, highlighting the extra time needed to load and run the reward model. Figure 4 presents the results, demonstrating that ST-BoN significantly reduces inference latency compared to Full-BoN. **Overall, wall-clock time latency drops by nearly 50%, with the reduction ratio growing as $N$ increases.** This efficiency stems from the

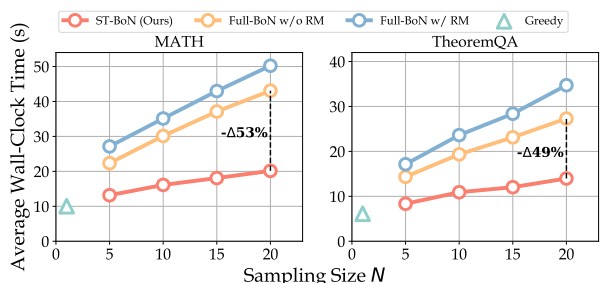

Figure 4: The average wall-clock time (seconds) comparisons in two datasets with different sampling sizes $N$.

self-truncating operation, which compensates for delays from self-estimation. Additionally, incorporating a reward model into Full-BoN further increases inference latency.

# 5 Experiments

## 5.1 Setup

**Datasets.** We select four datasets for objective tasks: MATH [18], TheoremQA [8], GPQA [37], and MMLU [17]. They span a range of domains, including mathematics, theorem application, science reasoning, and general knowledge, and present a significant difficulty level. We also select two datasets for subjective tasks: CNNDM [34] and AlpacaFarm [11]. The former is the summarization task about the open-ended generation, and the latter is the instruction-following task about the preference alignment. We adopt the revised CNNDM [51] for its higher-quality gold summaries.

**Models.** We mainly adopt 7B+ parameter models with the Zero-Shot-CoT generation paradigm [52, 23], including Qwen2.5-7B-Instruct [55], Llama3-8B-Instruct [10], and Mistral-7B-Instruct-v0.3 [22]. We also test the Qwen2.5-72B-Instruct [55] to validate generalization across model scales.

**Baselines.** We compare our ST-BoN decoding with two Full-BoN decoding paradigms: Full-BoN without a reward model (Full-BoN w/o RM) and Full-BoN with a reward model (Full-BoN w/ RM). In objective tasks: Full-BoN w/o RM uses the classic self-consistency through majority voting [47], and Full-BoN w/ RM employs the process reward model (PRM) Skywork-o1-Open-PRM-7B [2][3]. In subjective tasks: Full-BoN w/o RM applies the modified self-consistency through semantic similarity [21], and Full-BoN w/ RM uses the ArmoRM-Llama-3-8B [45] for preference rewards.

**Implementation.** We use the sampling strategy combining top-$k$ [13], top-$p$ [20], and temperature $T$ [19], with $k = 20$, $p = 0.95$, and $T = 0.7$. The buffer window length $\tau$ is set to $c$ in our main experiments. Detailed hyperparameter analysis is provided in Section 6.2 and 6.3. All baselines are implemented using the HuggingFace Transformers [43] library's `model.generate()` function with KV cache. All experiments are run on 80G A100 GPUs, with the GPU number varying based on $N$.

---

[2]`https://huggingface.co/Skywork/Skywork-o1-Open-PRM-Qwen-2.5-7B`

[3]We use PRMs that are as strong and generic as possible, selecting one of the most advanced for our main experiments, and Appendix D.2 includes results with other PRMs to validate the generality of our conclusions.

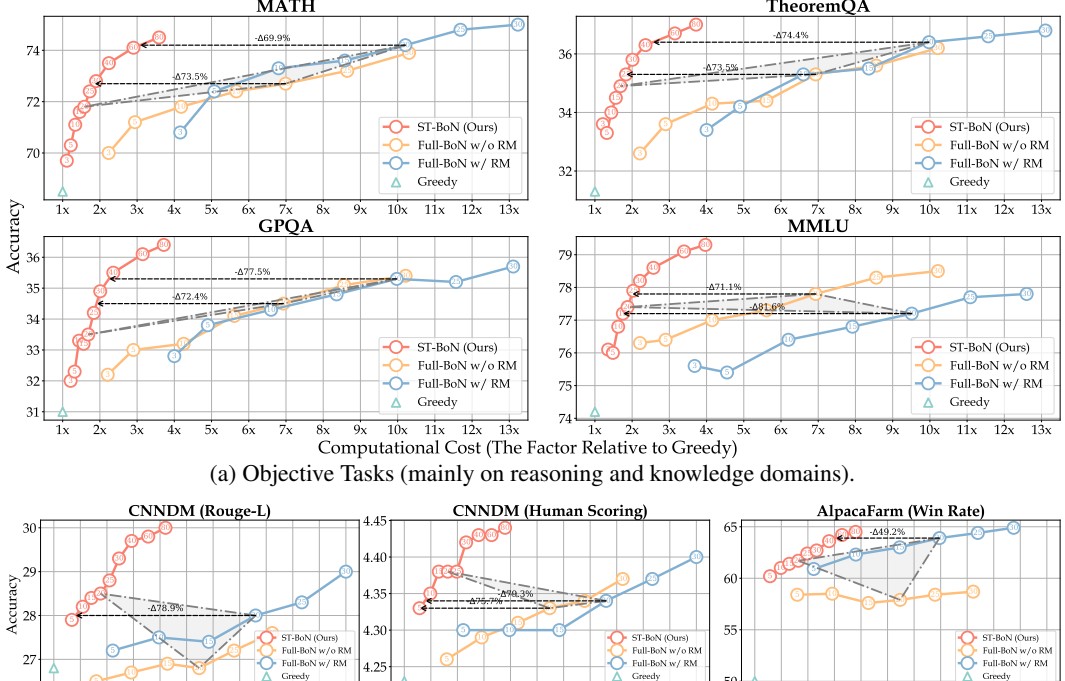

(a) Objective Tasks (mainly on reasoning and knowledge domains).

(b) Subjective Tasks (open-ended generation and preference alignment).

Figure 5: Computational cost and accuracy on different objective and subjective task datasets with various domains using the **Qwen2.5-7B-Instruct** model. Each point on the curves corresponds to a specific sampling size $N$ from 3 to 80, with values labeled within the circle. We connect the three results at $N = 20$ with gray dashed lines to show their relative performance without considering cost.

**Evaluation.** We mainly evaluate **the balance between computational cost and performance**. The computational cost lies in two dimensions in Section 4: memory and time. We use the cost of greedy decoding as the baseline, and the cost of each paradigm is calculated as follows:

- *Memory Cost* $\mathcal{M}_{\text{cost}}$: Let $M_{\text{bm}}$, $M_{\text{peak}}$, and $M_{\text{rm}}$ denote the memory usage of the base LLM weights, the peak memory overhead during inference, and the reward model weights, respectively, for each paradigm. Let $M_{\text{peak}}^{\text{greedy}}$ denote the peak memory during greedy decoding:

$$\mathcal{M}_{\text{cost}} := \begin{cases} (M_{\text{bm}} + M_{\text{peak}}) \, / \, (M_{\text{bm}} + M_{\text{peak}}^{\text{greedy}}), & \text{ST-BoN \& Full-BoN w/o RM,} \\ (M_{\text{bm}} + M_{\text{peak}} + M_{\text{rm}}) \, / \, (M_{\text{bm}} + M_{\text{peak}}^{\text{greedy}}), & \text{Full-BoN w/ RM.} \end{cases} \tag{8}$$

- *Time Cost* $\mathcal{T}_{\text{cost}}$: $\mathcal{T}_{\text{cost}}$ is denoted as the ratio of the wall-clock time of each paradigm to that of greedy decoding. For Full-BoN w/ RM, we consider that both the generator and the verifier are persistently stored in GPU memory, and thus ignore the loading time of the RM.

The **overall computational cost** $\mathcal{A}_{\text{cost}}$ is given by:

$$\mathcal{A}_{\text{cost}} := \mathcal{M}_{\text{cost}} \cdot \mathcal{T}_{\text{cost}}. \tag{9}$$

Regarding **performance**, *Accuracy* is used as a metric in reasoning scenarios. To reduce random error, we follow [16] to report the average results over four evaluation runs. In open-ended scenarios, *Rouge-L* [29] and *Human Scoring* are applied for the summarization, and *Win Rate (WR)* [59] judged by GPT-4o-Turbo [1] is used for the instruction-following. Metric details are shown in Appendix C.

## 5.2 Results I: Objective Task

Figure 5a presents the results of Qwen2.5-7B-Instruct on four objective tasks. Due to space constraints, results for other models are provided in Appendix D.1. First, we analyze the *result soundness*. Under the same sampling size $N$ (we use $N = 20$ as an example and connect the three paradigms with gray dashed lines), when cost is ignored, compared to Full-BoN w/o RM, ST-BoN shows no significant performance drop, indicating that **the theory proposed in Section 3.1** — early consistency

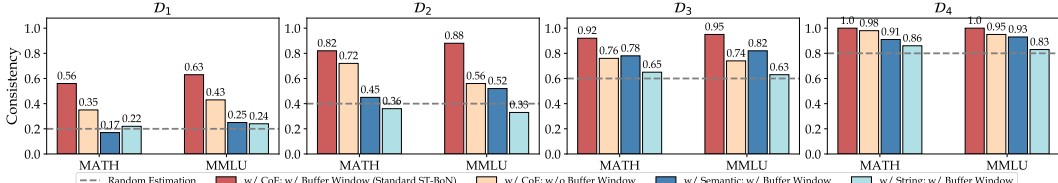

Figure 6: The early-final consistencies between early self-estimation and final correctness in different domains and sub-datasets with the **Qwen2.5-7B-Instruct** model. $\mathcal{D}_i$ represents the subset containing all cases from dataset $\mathcal{D}$ that produces $i$ correct answers out of $N$ samplings.

foreshadows final consistency — **is reasonable and holds in practice.** In addition, Full-BoN w/ RM outperforms the others in three datasets more related to reasoning (except MMLU), which aligns with the test-time scaling goal of "increasing inference cost to improve performance" [40].

Next, we analyze our core advantage: ***cost-performance trade-offs***. Due to self-truncation, ST-BoN does not fully utilize all $N$ samples, leading to lower performance than Full-BoN initially. However, it greatly reduces the cost, allowing $N$ to be scaled up within the saved cost. As $N$ increases, ST-BoN eventually matches Full-BoN's performance while using lower costs. For example, when Full-BoN uses $N = 20$, ST-BoN reaches the same accuracy at $N = 20 \sim 30$ with 70–75% lower costs (vs. Full-BoN w/o RM), and at $N = 40 \sim 80$ with 75–80% lower costs (vs. Full-BoN w/ RM), showing that **ST-BoN can achieve the same performance as Full-BoN with significantly lower costs**.

On the other hand, under the same computational cost, ST-BoN can scale to a much larger $N$ than Full-BoN. For example, when Full-BoN w/o RM uses $N = 10$ and Full-BoN w/ RM uses $N = 3$, ST-BoN can reach nearly $N = 80$. This gives it access to much more samples, leading to a 3-4 point higher accuracy across all datasets. These results show that **under the same computational cost, ST-BoN can achieve better performance than Full-BoN**.

Finally, we also observe the ***domain generalization*** of ST-BoN. Full-BoN w/o RM performs noticeably worse on MMLU, a knowledge dataset, compared to the other three reasoning datasets. In contrast, ST-BoN maintains its cost–performance trade-off advantage. This suggests that Full-BoN w/o RM relies on a strong and general RM to be effective, while ST-BoN avoids this potential cost. **By not requiring domain-specific priors, ST-BoN remains effective across diverse domains**.

### 5.3 Results II: Subjective Tasks

To evaluate the domain generalization of ST-BoN, we further test it on subjective tasks. The results for Qwen2.5-7B-Instruct are shown in Figure 5b, with results of other models shown in Appendix D.1. First, Full-BoN w/o RM performs poorly, indicating that semantic consistency cannot identify optimal samples. Second, Full-BoN w/ RM underperforms on summarization tasks due to a domain mismatch: the reward model is typically trained on open-domain QA preferences, which rarely cover summarization, leading to OOD issues. Finally, on instruction-following tasks, although Full-BoN w/ RM outperforms ST-BoN at the same sampling size $N$, ST-BoN can reach its performance by increasing $N$, while reducing computational cost by approximately 50%. **This highlights ST-BoN's cost-performance trade-off on subjective tasks and further demonstrates its domain robustness.**

## 6 In-depth Analysis

### 6.1 Component Ablation: Why ST-BoN Works?

Our method is motivated by the idea that early consistency may foreshadow final consistency, then we use CoE to self-estimate early consistency, and propose a buffer window to mitigate noise. In this part, we will ablate the two components: CoE measure and buffer window, to evaluate their necessity.

**Baseline.** The standard ST-BoN involves "w/ CoE & w/ Buffer Window". First, we ***remove the buffer window***, allowing the LLM to perform only one self-estimation at the earliest estimation time $c$. Next, to further assess the necessity of using hidden states of LLMs (*i.e.*, CoE) for early consistency computation, we compare two unsupervised output measure: ***semantic*** and ***string***. Specifically, in Eq.5, we replace $\mathcal{D}(Y_{1:c}^i, Y_{1:c}^j)$ with $||[\boldsymbol{h}_{1:c}^i]_L - [\boldsymbol{h}_{1:c}^j]_L||_2^2$ and $1/\text{Rouge-L}(Y_{1:c}^i, Y_{1:c}^j)$, respectively.

**Evaluation.** We assess the ***Early-Final Consistency*** between *early self-estimation* and *final correctness*. Specifically, assume each case in dataset $\mathcal{D}$ is sampled $N$ times. The dataset is divided

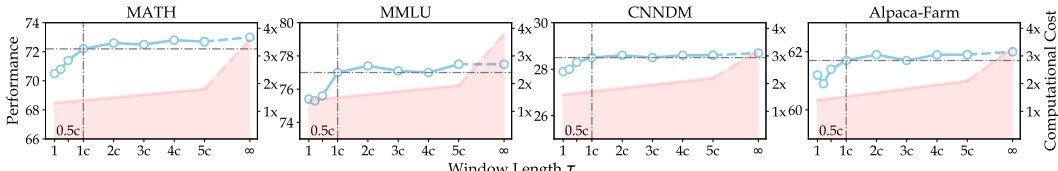

Figure 7: Hyperparameter ablation with varying window length $\tau$. The blue line shows performance changes as $\tau$ increases, while the pink shading represents changes in computational cost. $\tau \to \infty$ indicates a window extended until any sampling reaches [EOS].

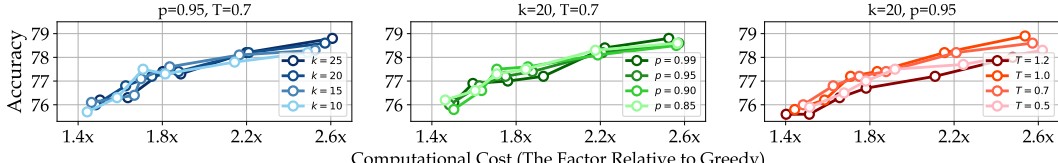

Figure 8: Sampling strategy ablation with various top-$k$, top-$p$, and temperature $T$ in MMLU.

into $N + 1$ disjoint subsets $\{\mathcal{D}_i\}_{i=0}^N$, where $\mathcal{D}_i$ contains cases that yield $i$ correct answers out of $N$ samples. For each $\mathcal{D}_i$, we compute the proportion of cases where the LLM's self-estimated best sampling produces a correct final answer, representing early-final consistency. Naturally, as $i$ decreases, the estimation becomes more challenging, and the random consistency expectation is $i/N$.

**Results.** We evaluate on the MATH and MMLU datasets due to their diverse domains and difficulty levels. With $N = 5$, we exclude subsets $\mathcal{D}_0$ and $\mathcal{D}_5$, as they represent cases that are entirely correct or incorrect. Figure 6 presents the results using the Qwen2.5-7B-Instruct model (Results for other models are in Appendix D.3), we find that: **(i)** "w/ buffer window" consistently outperforms that "w/o buffer window", demonstrating its effectiveness. **(ii)** For all subsets, "w/ CoE" significantly outperforms random estimation, with standard ST-BoN achieving excellent consistency. In contrast, "w/ semantic" and "w/ string" hover around the random baseline, highlighting the critical role of CoE in early consistency measure. **Thus, each module in the current ST-BoN plays an essential role.**

In fact, we have also discovered that **using CoE consistency under full generation, yields better results than using self-consistency with majority voting**. This provides further evidence that CoE's effectiveness in measuring early consistency is not random. The results are shown in Appendix D.4.

## 6.2 Window Length $\tau$ Ablation

In ST-BoN, the buffer window length $\tau$ is the only hyperparameter. We study its impact by varying $\tau$ in different datasets with Qwen2.5-7B-Instruct, with a fixed sampling size $N = 10$.

Figure 7 shows that when $\tau < c$, the performance gain relative to cost increases faster. However, beyond $c$, this gain slows significantly, making $c$ the optimal choice for balancing performance and cost in our experiments. Furthermore, if ignoring costs, performance continues to show fluctuating improvements as $\tau$ increases, suggesting the potential of task-specific $\tau$ tuning. For example, on the MATH dataset, performance improves more steadily with larger $\tau$ values, probably due to its longer outputs, where early estimates are less informative, and need a larger buffer window to capture more information. Adaptive $\tau$ based on task complexity can be a promising direction for future work.

## 6.3 Sampling Strategy Robustness

We also evaluate the robustness of ST-BoN across different sampling strategies by varying top-$k$, top-$p$, and temperature $T$ settings, and analyze their impact on performance and computational cost.

Figure 8 shows the results on the MMLU dataset using the Qwen2.5-7B-Instruct model. We observe that variations in $k$ and $p$ have a minimal impact on performance and cost, while changes in $T$ affect both more significantly. Unlike $k$ and $p$, which only shift the truncation point of the probability distribution without affecting token relativity, $T$ alters token probabilities, making the distribution more uniform and increasing sampling randomness as $T$ increases. This results in an earlier occurrence of the earliest estimation time $c$, thereby reducing computational cost. However, an earlier estimation time $c$ might capture less effective information, potentially as a side effect of improved efficiency. We recommend using a moderate $T$ value, such as 0.7, to strike a balance between performance and cost.

### 6.4 Case Study

We also conduct case studies for further comparison. We find that ST-BoN can better address ambiguous scenarios, such as when sampling answers are all inconsistent or majority voting encounters high randomness, compensating for these limitations. Details are available in Appendix E.

## 7 Related Work and Comparisons

### 7.1 Test-Time Scaling

BoN aims to achieve test-time scaling [40]. The idea is to increase the compute during inference while setting cost aside, to push the performance ceiling. Existing approaches fall into two broad lines [32]: *Sequential scaling* lengthens a single reasoning path during generation and seeks deeper deliberation, reflection, or insight [16]. *Parallel scaling*, which captures the spirit of BoN, increases the number of reasoning paths generated at once and seeks greater diversity in reasoning [47, 6]. Other methods, such as ToT [57], GoT [5], and Monte Carlo Tree Search [44], share the same objective but expand from a reasoning-structure perspective, taking different research tracks than BoN.

### 7.2 Best-of-$N$ Sampling

BoN sampling was originally proposed for inference-time preference alignment [33], using a reward model to select the best sample and further fine-tune the base model [42]. Its effectiveness has been theoretically supported by several studies [12, 31, 4, 56]. BoN has also seen broad adoption in reasoning tasks [40], notably in the self-consistency paradigm [47], which uses majority voting to select answers unsupervisedly. More recently, process reward models [28, 46] have enabled direct scoring within BoN for reasoning. However, reward models raise computational costs and are vulnerable to overoptimization, and rely heavily on their model capabilities [14].

### 7.3 Efficient Best-of-$N$

The core optimization idea of ST-BoN can be summarized as ***"depth pruning"***, which eliminates some suboptimal samples in early decoding to reduce memory usage. SBoN [41] also considers removing local segments; CARDS [25] and TreeBoN [36] continue to branch after discarding them. However, these methods still rely on a reward model for scoring, so the additional inference latency and memory usage remain high. More importantly, they have been validated only under the preference alignment, but in reasoning domains, reward models may be difficult to score randomly segmented process fragments. **Appendix B offers a detailed comparison, highlighting the greater efficiency of ST-BoN and its broader applicability in different domains.**

Another type of optimization idea can be summarized as ***"breadth pruning"***, aiming to adaptively reduce the actual sampling number under a fixed budget $N$. For example, ASC [2] stops sampling early based on answer frequency, and ESC [27] dynamically adjusts the required sampling sizes from an entropy perspective. Although these methods lower token output and reduce memory usage, they rely on serial sampling with adaptive stopping, which compromises the parallelism benefits of BoN and limits efficiency on modern GPUs. As a result, they are suboptimal in terms of inference latency.

Certainly, another research line studies distilling the BoN policy during training to replace multiple samplings with a single one, reducing inference-time latency. Methods like BoNBoN [15], BOND [39], and vBoN [3] focus on this line, shifting the computational cost from the inference phase to the training phase. Overall, this line explores more about the potential of learning the BoN distribution at training time, making orthogonal contributions compared to direct inference-time optimization.

## 8 Conclusion

We propose ST-BoN decoding, which enables LLMs to self-estimate the most promising sampling without fully generating $N$ samples or using reward models. ST-BoN significantly reduces GPU memory overhead and inference latency while demonstrating better cost-performance trade-offs than Full-BoN in both objective and subjective tasks from reasoning to preference alignment.

## Limitations

- **(1) Model Transparency:** Since accessing the hidden states of the LLM is required, ST-BoN does not apply to closed-source models such as OpenAI's GPT-4 series [1]. However, with the rapid development of open-source LLMs [16], we believe that research on white-box methods is crucial, as it can provide stronger interpretability and help us better explore the internal mechanisms of LLMs.

- **(2) Length Adaptivity:** As mentioned in Section 6.2, there is still room for improvement in ST-BoN's ability to adaptively adjust the window length $\tau$ based on task complexity. Longer responses may require longer windows to capture more information. To better balance the cost, we can observe the average generation length of the task during greedy decoding and adjust the window size accordingly. This ensures that performance is maximized within an acceptable cost range.

## Societal Impact

- **(1) Trustworthiness:** ST-BoN relies on internal consistency signals to guide decoding, moving the selection process into the model's latent space. Prior work [47] has shown a strong correlation between consistency and correctness. Building on this, our theoretical analysis (Section 3.1) and empirical findings (Section 6.1) further support the reliability of latent-space decision-making. Together, these two parts provide a solid foundation for the model's trustworthiness under our method.

- **(2) Safety:** ST-BoN introduces no additional safety risks. The sampling follows the model's native distribution without adversarial perturbation, and the consistency-based selection mechanism operates without external reward models, thus avoiding interference from out-of-distribution signals.

- **(3) Biases:** ST-BoN inherits the base model's biases, as it does not perform explicit debiasing. However, by leveraging multiple parallel samplings, it can identify and truncate transient biases that arise sporadically during generation. These biased trajectories may be excluded as outliers through the consistency-based strategy, thereby potentially mitigating their impact.

## Acknowledgement

This paper was supported by the General Program of the National Natural Science Foundation of China (62176153). This paper was also supported by the National Natural Science Foundation of China (62406188) and the Natural Science Foundation of Shanghai (24ZR1440300). Additionally, this work was partially conducted during Yiming's internship at the Tongyi Lab, Alibaba Group. It was supported by the Alibaba Research Intern Program.

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

# Appendix

## A   Proof of Theorem 1

*Proof.* We first make the two key structural assumptions:

- Local Lipschitz continuity of the model: The model's next-token distributions satisfy, for all partial sequence pairs $(Y_{1:t}^i, Y_{1:t}^j)$,

$$\left\| P_{t+1}(\cdot \mid Y_{1:t}^i) - P_{t+1}(\cdot \mid Y_{1:t}^j) \right\|_{\mathrm{TV}} \le L \cdot \mathcal{D}(Y_{1:t}^i, Y_{1:t}^j), \tag{10}$$

  for some $L \ge 0$, where $\| \cdot \|_{\mathrm{TV}}$ denotes the Total Variation Distance. It quantifies the "smooth" behavior that tiny changes to the sequence produce only tiny shifts in the next-token probabilities. Transformers are inherently Lipschitz-continuous: Each token embedding passes through layers of self-attention and feed-forward networks whose composition preserves this smoothness, so similar sequences yield similar hidden states. Moreover, Softmax and LayerNorm further suppress perturbations, preventing small input changes from causing large changes in the output.

- Bounded-increment of the distance: The distance measure $\mathcal{D}$ satisfies: for any two partial sequences $Y_{1:t}^i, Y_{1:t}^j$ and any next tokens $Y_{t+1}^i, Y_{t+1}^j$,

$$\mathcal{D}(Y_{1:t}^i \circ Y_{t+1}^i, \ Y_{1:t}^j \circ Y_{t+1}^j) \le \mathcal{D}(Y_{1:t}^i, Y_{1:t}^j) + M \cdot \mathbf{1}[Y_{t+1}^i \ne Y_{t+1}^j], \tag{11}$$

  for some constant $M > 0$. It ensures the distance measure remains controlled, preventing abrupt shifts from a single token change. Fundamentally, this reflects the smoothness characteristic of local Lipschitz continuity.

Next, we start our proof. Consider the pair $(Y_{1:t}^1, Y_{1:t}^i)$. By the maximal coupling of $P_{t+1}(\cdot \mid Y_{1:t}^1)$ and $P_{t+1}(\cdot \mid Y_{1:t}^i)$, The probability they draw different tokens at step $t+1$ is

$$\Pr\left[Y_{t+1}^1 \ne Y_{t+1}^i \mid d_t^i\right] = \left\| P_{t+1}(\cdot \mid Y_{1:t}^1) - P_{t+1}(\cdot \mid Y_{1:t}^i) \right\|_{\mathrm{TV}} \le L\, d_t^i. \tag{12}$$

If they differ, the bounded-increment assumption implies

$$d_{t+1}^i - d_t^i \le M. \tag{13}$$

Hence

$$\begin{aligned}
\mathbb{E}[d_{t+1}^i \mid d_t^i] &= \mathbb{E}[d_{t+1}^i - d_t^i + d_t^i \mid d_t^i] \\
&\le d_t^i + \Pr[Y_{t+1}^1 \ne Y_{t+1}^i \mid d_t^i]\, M \\
&\le d_t^i + (L\, d_t^i)\, M = (1 + LM)\, d_t^i.
\end{aligned} \tag{14}$$

Let $\Gamma = 1 + LM$. Summing the above over $i = 2, \dots, N$ gives

$$\mathbb{E}[S_{t+1} \mid S_t] = \sum_{i=2}^{N} \mathbb{E}[d_{t+1}^i \mid d_t^i] \le \Gamma \sum_{i=2}^{N} d_t^i = \Gamma\, S_t. \tag{15}$$

Starting from $t$ and iterating the one-step bound,

$$\begin{aligned}
\mathbb{E}[S_{t+1}] &\le \Gamma\, S_t, \\
\mathbb{E}[S_{t+2}] &\le \Gamma\, \mathbb{E}[S_{t+1}] \le \Gamma^2 S_t, \\
&\ \ \vdots \\
\mathbb{E}[S_T] &\le \Gamma^{T-t} S_t.
\end{aligned} \tag{16}$$

For any $\epsilon > 0$, Markov's inequality yields

$$\Pr[S_T \ge \epsilon \mid S_t] \le \frac{\mathbb{E}[S_T \mid S_t]}{\epsilon} \le \frac{\Gamma^{T-t}\, S_t}{\epsilon}. \tag{17}$$

Rearranging gives the stated result in Theorem 1:

$$\Pr[S_T \le \epsilon \mid S_t] \ge 1 - \frac{\Gamma^{T-t}}{\epsilon} S_t. \tag{18}$$

$\square$

# B   Related Work Comparison

Some existing work also considers the reduction of the full generation to improve the efficiency of BoN. The core idea behind all three methods is to use the reward model to score partial sequences at certain time steps during the sampling process. SBoN sets a threshold and discards all samples with scores above this threshold, then iteratively generates and selects the remaining candidates until only one remains. TreeBoN and CARDS retain only one candidate at each selection and then repeatedly branch out $N$ new paths, performing selection at each branching stage. These methods have fundamental differences from ST-BoN. Table 1 presents a comparison between different dimensions.

Table 1: Detailed comparison with other efficient BoN methods aiming to reducing full generation.

| Method | Free of Domain Prior | Free of Reward Models | Memory Bottleneck (excluding KV cache) | Time Bottleneck | Applicable Domain |
|---|---|---|---|---|---|
| SBoN [41] | ✗ | ✗ | | | |
| CARDS [25] | ✗ | ✗ | Reward Model Weight | Load & Run Reward Models | Preference Alignment |
| TreeBoN [36] | ✓ | ✗ | | | |
| ST-BoN (Ours) | ✓ | ✓ | N/A | Vector Operations | All Objective Tasks (e.g. Reasoning & Knowledge) and Subjective Tasks (e.g. Generation & Preference Alignment) |

From the perspective of computational cost, the memory overhead and inference time introduced by the reward model are non-negligible. Additionally, during the autoregressive generation process, the reward model repeatedly scores incomplete segments, which blocks parallel sampling and significantly reduces actual GPU utilization. Therefore, in practical production settings, from a wall-clock-time perspective, these approaches still face efficiency bottlenecks.

On the other hand, the effectiveness of SBoN and CARDS relies on a prior assumption: the reward model's score for a sampling prefix correlates with the score for the final complete text. However, this assumption has only been validated in preference alignment domains, raising concerns about domain generalizability. When applied to reasoning domains, process reward models will likely struggle to evaluate arbitrarily segmented text, as they tend to focus on scoring complete process segments. Furthermore, [36] has pointed out that the prior assumption underlying CARDS breaks down when the generation length exceeds 128 tokens. **In contrast, ST-BoN is highly lightweight: it does not incur overhead or blocking from a reward model and does not depend on domain-specific priors.**

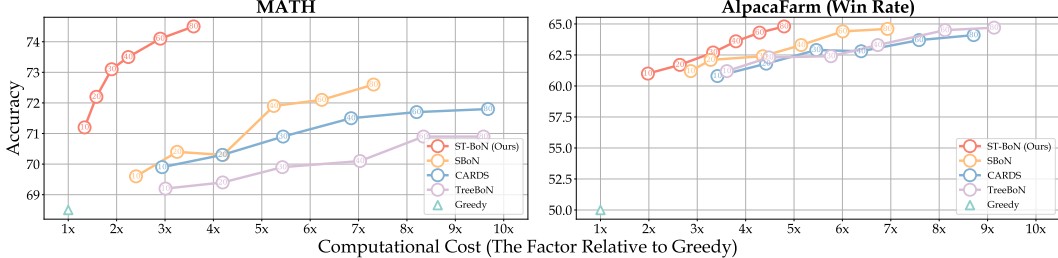

Figure 9: Computational cost and accuracy on different objective and subjective task datasets with various domains using the **Qwen2.5-7B-Instruct** model. Each point on the curves corresponds to a specific sampling size $N$ from 10 to 80, with values labeled within the circle. **We compare our ST-BoN with three other efficient BoN methods.**

We compare the cost-performance trade-offs of ST-BoN and the three methods on both an objective and a subjective task, as illustrated in Figure 9. (i) The MATH benchmark evaluates the reasoning ability. All three other methods perform poorly on this task, probably because process reward models require complete procedural segments to assign reliable scores. They struggle to make sound decisions when evaluating arbitrarily segmented text fragments. (ii) The AlpacaFarm benchmark assesses preference alignment, the primary domain where the three methods have demonstrated effectiveness. In this domain, ST-BoN performs comparably or even better than the others. This may be because the inductive biases assumed by some methods do not generalize well across different models. These results collectively demonstrate the strong domain generalizability and practical applicability of ST-BoN.

## C  Evaluation Setup

- **Reasoning Scenarios**: We use *Accuracy* as the evaluation metric. Following [47], we extract answers from LLM responses using the exact match method and compare them with the ground truth.

- **Open-end Scenarios**:
  - **Summarization**: We use both automatic and human evaluation. For the automatic metric, we adopt *Rouge-L* [29]. For human evaluation, volunteers score LLM-generated summaries across four dimensions [24, 51]: (i) Fluency: Free of spelling, grammatical, or syntactic errors; (ii) Coherence: Events should be logically connected, with smooth linguistic transitions; (iii) Consistency: No hallucinated facts; all information must align with the source document; (iv) Relevance: Emphasizes key information while minimizing non-core facts and redundant details. Each dimension is rated on a scale of 0 to 5, and the final score is the average of the four dimension scores. We invited three graduate students to individually score 50 samples from the dataset following [51]. The final reported score is the average of the three annotations. Each annotator spent approximately 16 hours on average to complete the task.
  - **Instruction-following**: We use the BoN alignment metric, *Win Rate (WR)* [59]. Each paradigm's responses were paired with those generated via greedy decoding and assessed by an external LLM, GPT-4-Turbo [1]. The judge model selects the better response in each pair, and WR is computed as the percentage of comparisons where the paradigm's response is preferred. Greedy decoding serves as a baseline with a WR of 50%, representing random performance.

## D  Supplementary Experimental Results

### D.1  Main Results of All LLMs

In this part, we present the main experimental results for all remaining LLMs, including: Qwen2.5-72B-Instruct (Figure 10), Llama3-8B-Instruct (Figure 11), and Mistral-7B-v0.3-Instruct (Figure 12). As highlighted in [46], a significant performance drop occurs when the verifier's parameter scale is smaller than that of the generator. Therefore, for the Qwen2.5-72B-Instruct model, we adopt a PRM of equal scale — Qwen2.5-Math-PRM-72B [58], and only report results in objective tasks.

Under these models, all conclusions regarding the cost–performance trade-off drawn in Sections 5.2 and 5.3 still hold, and as model size increases, ST-BoN remains effective, demonstrating its robustness to parameter scaling.

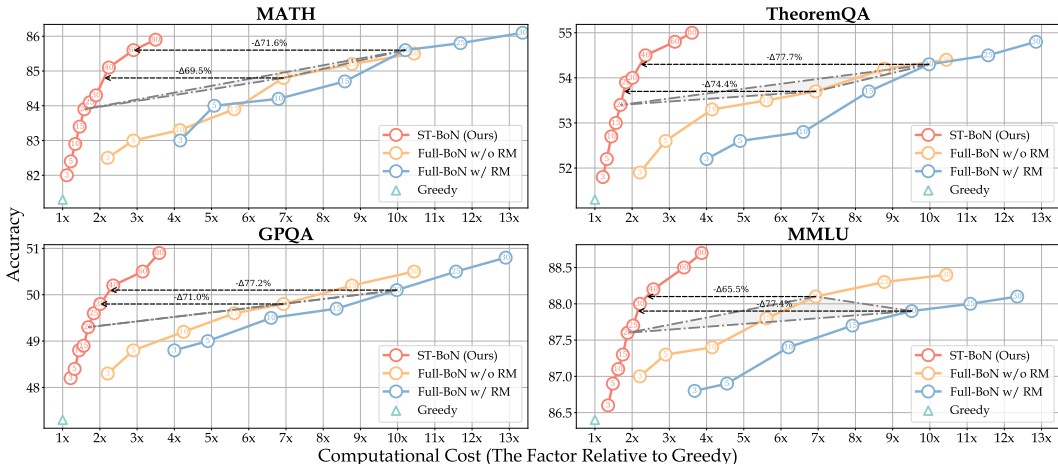

Figure 10: Computational cost and accuracy on different objective task datasets with various domains using the **Qwen2.5-72B-Instruct** model. Each point on the curves corresponds to a specific sampling size $N$ from 3 to 80, with values labeled within the circle. We connect the three results at $N = 20$ with gray dashed lines to show their relative performance without considering cost.

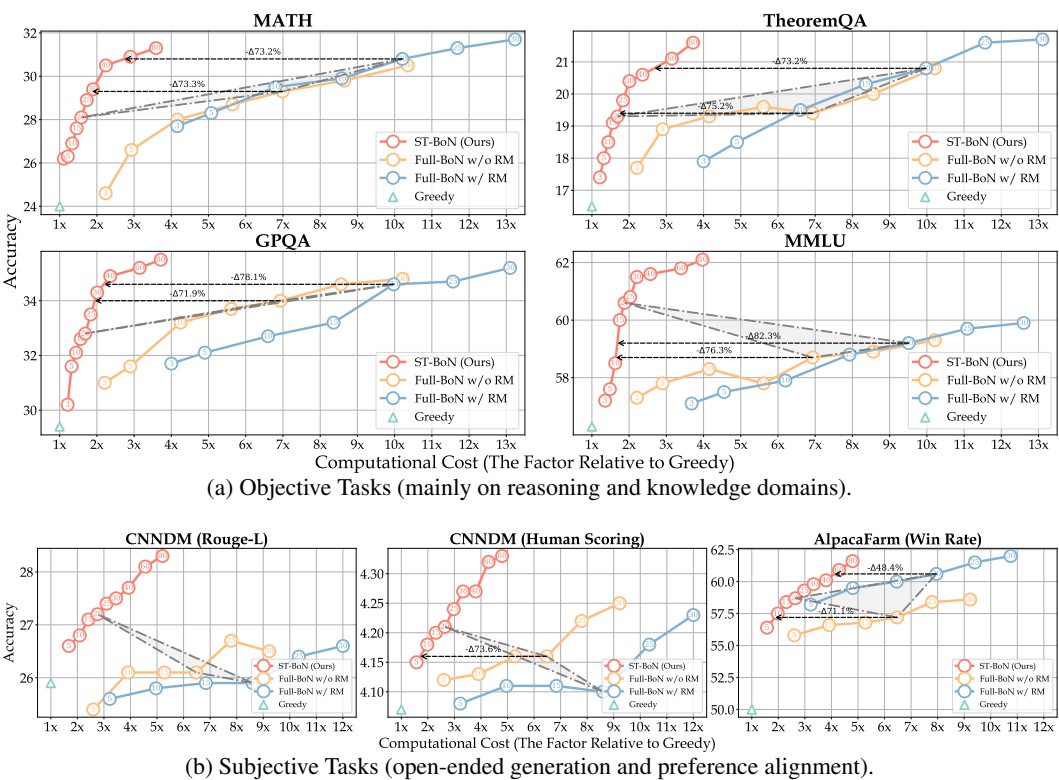

(a) Objective Tasks (mainly on reasoning and knowledge domains).

(b) Subjective Tasks (open-ended generation and preference alignment).

Figure 11: Computational cost and accuracy on different objective and subjective task datasets with various domains using the **Llama-3-8B-Instruct** model. Each point on the curves corresponds to a specific sampling size $N$ from 3 to 80, with values labeled within the circle. We connect the three results at $N = 20$ with gray dashed lines to show their relative performance without considering cost.

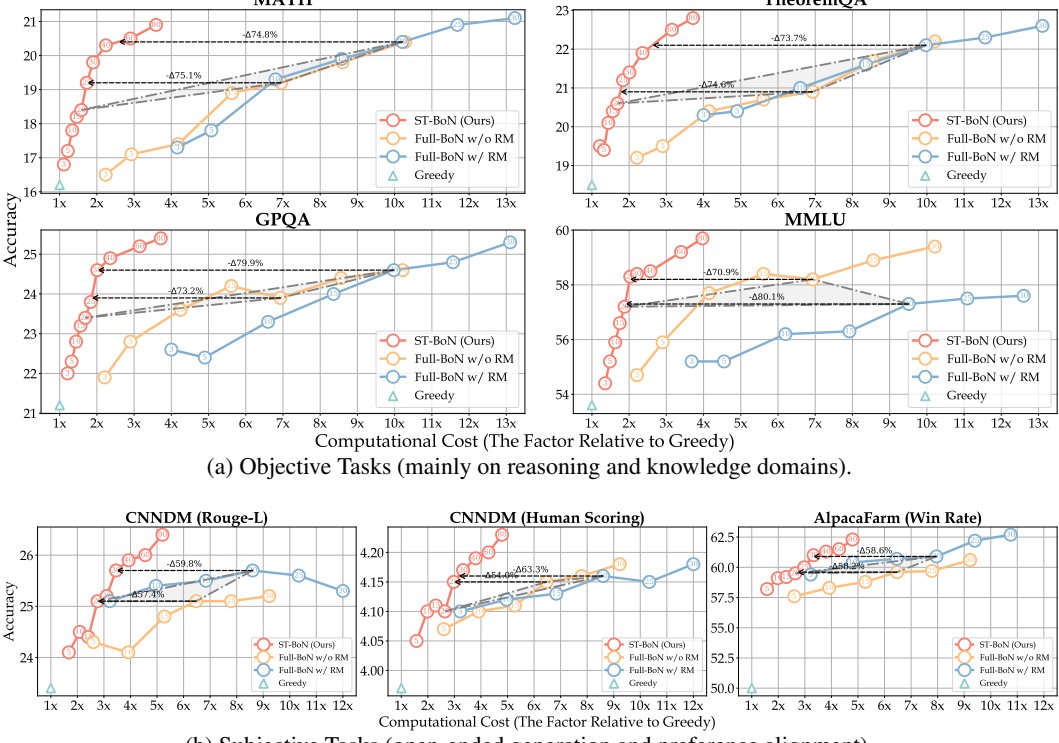

(a) Objective Tasks (mainly on reasoning and knowledge domains).

(b) Subjective Tasks (open-ended generation and preference alignment).

Figure 12: Computational cost and accuracy on different objective and subjective task datasets with various domains using the **Mistral-7B-Instruct-v0.3** model. Each point on the curves corresponds to a specific sampling size $N$ from 3 to 80, with values labeled within the circle. We connect the three results at $N = 20$ with gray dashed lines to show their relative performance without considering cost.

## D.2  Reward Model Ablation

To ensure that the choice of reward models does not impact our key conclusions, we extend the baselines of the Full-BoN w/ RM paradigm beyond the main experiment in Figure 13. Specifically, we introduce two additional advanced reward models for comparisons: Qwen2.5-Math-PRM-7B [58] and RLHFlow-PRM-Mistral-8B [54]. As shown in Figure 13, **the choice of reward models does not affect the conclusions regarding the cost-performance trade-off advantages of ST-BoN in the main experiments.**

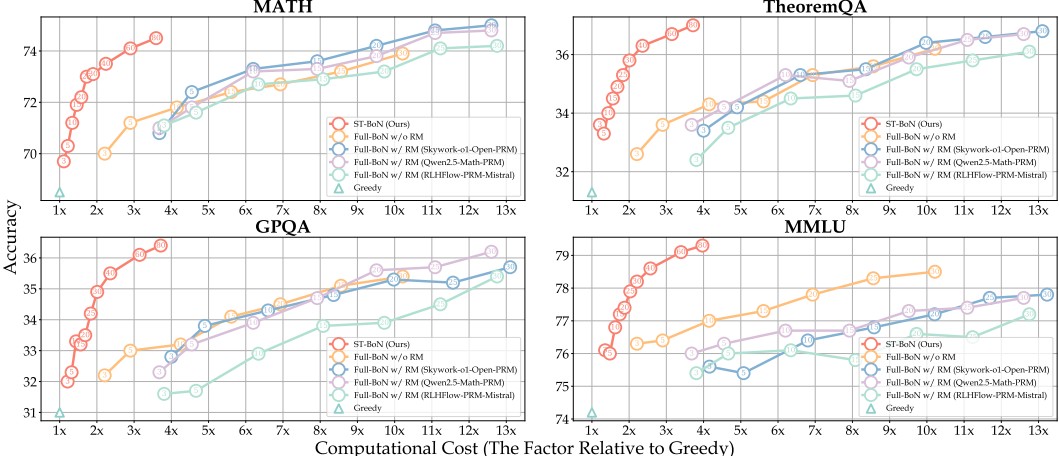

Figure 13: **(Reward Model Ablation)** Computational cost and accuracy on different objective and subjective task datasets with various domains using the **Qwen2.5-7B-Instruct** model. Each point on the curves corresponds to a specific sampling size $N$ from 3 to 80, with values labeled within the circle. **For the Full-BoN w/ RM paradigm, we use three reward models for comparisons.**

## D.3  Component Ablation of All LLMs

We present the additional early-final consistency results on Llama3-8B-Instruct (Figure 14) and Mistral-7B-Instruct-v0.3 (Figure 15) models, demonstrating a strong positive correlation between ST-BoN's early self-estimation and the final answer correctness, as concluded in Section 6.1. This indicates that our consistency conclusion can be generalized across various series of LLMs.

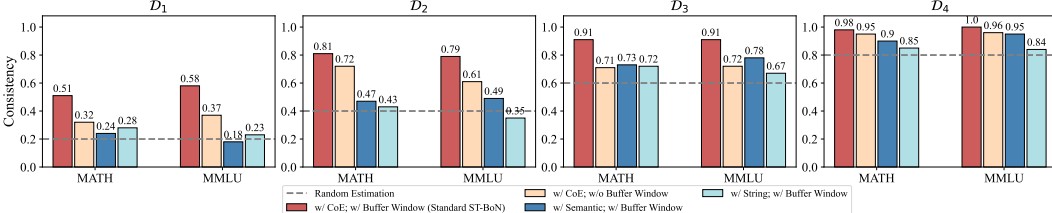

Figure 14: The early-final consistencies between early self-estimation and final correctness in different domains and sub-datasets with the **Llama3-8B-Instruct** model. $\mathcal{D}_i$ represents the subset containing all cases from dataset $\mathcal{D}$ that produces $i$ correct answers out of $N$ samplings.

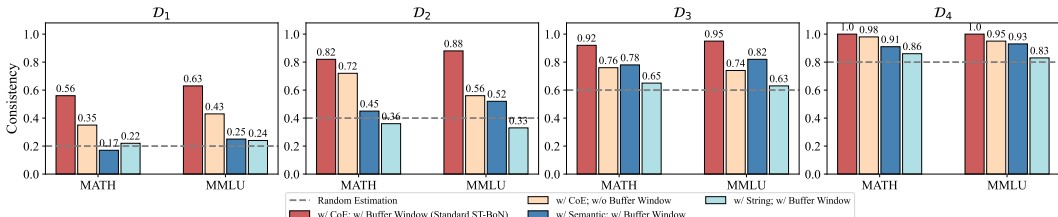

Figure 15: The early-final consistencies between early self-estimation and final correctness in different domains and sub-datasets with the **Mistral-7B-Instruct-v0.3** model. $\mathcal{D}_i$ represents the subset containing all cases from dataset $\mathcal{D}$ that produces $i$ correct answers out of $N$ samplings.

### D.4 Full-BoN with Chain-of-Embedding

In our experiments, we find that under the same sampling size $N$, although the absolute performance of ST-BoN was weaker than that of Full-BoN w/o RM, the gap is not substantial. Since our self-estimation of early consistency relies on the CoE measure, this raises a further question regarding the effectiveness of CoE: *Would using CoE in Full-BoN w/o RM outperform majority voting?*

Table 2: The performance comparisons of majority voting and chain-of-embedding under the Full-BoN w/o RM paradigm. We use the **Qwen2.5-7B-Instruct** model to evaluate two objective tasks.

| | $N=3$ | $N=5$ | $N=10$ | $N=15$ | $N=20$ | $N=25$ | $N=30$ | $N=40$ | $N=60$ | $N=80$ |
|---|---|---|---|---|---|---|---|---|---|---|
| **MATH** | | | | | | | | | | |
| Majority Voting [47] | **70.0** | **71.2** | 71.8 | 72.4 | 72.7 | 73.2 | **73.9** | 74.2 | 74.4 | 74.5 |
| Chain-of-Embedding [49] | 69.6 | 70.8 | **72.3** | **72.7** | **73.1** | **73.5** | 73.8 | **74.5** | **74.7** | **74.9** |
| **MMLU** | | | | | | | | | | |
| Majority Voting [47] | **76.3** | 76.4 | 77.0 | 77.3 | 77.8 | 78.3 | **78.5** | 78.6 | 78.6 | 79.2 |
| Chain-of-Embedding [49] | 76.2 | **76.5** | **77.5** | **77.8** | **78.2** | **78.5** | 78.4 | **78.9** | **79.2** | **79.5** |

Table 2 presents the performances of the Qwen2.5-7B-Instruct model on MATH and MMLU under the Full-BoN w/o RM paradigm, using majority voting [47] and CoE [49], respectively. Since the computational cost is nearly identical, we do not compare them in this regard. We observe that in most cases, using CoE outperforms majority voting. We think that majority voting focuses solely on the final answer, representing outcome supervision. However, LLMs might reach a correct answer through wrong reasoning, so the outcome doesn't always reflect the process information. Conversely, CoE utilizes each token to provide process supervision, which introduces more information than individual results. This indicates that the effectiveness of ST-BoN largely depends on the fact that **CoE is also a strong metric for selecting optimal samples under the Full-BoN paradigm**.

## E  Case Study

We present three cases as follows:

### • GPQA dataset in Qwen2.5-7B-Instruct model.

We set the sampling size $N=5$, with all sampling results and selected answers with ST-BoN and Full-BoN paradigms shown in Tables 3 - 6. The question is as follows:

> **Instruction:** Answer the following multiple choice question. The last line of your response should be of the following format: 'Answer: $LETTER' (without quotes) where LETTER is one of ABCD. Think step by step before answering.
>
> **Question:** We would like to dissolve (at 25°C) 0.1g Fe(OH)3 in 100 cm3 total volume. What is the minimum volume (cm3) of a 0.1 M monobasic strong acid that is needed to prepare the solution and what is the pH of the resulting solution?
> Choices:
> (A) pH 3.16; 32.14 cm3
> (B) pH 2.69; 30.09 cm3
> (C) pH 2.04; 28.05 cm3
> (D) pH 4.94; 20.40 cm3

### • TheoremQA dataset in Llama3-8B-Instruct model.

We set sampling size $N=5$, with all sampling results and selected answers with ST-BoN and Full-BoN paradigms shown in Tables 7 - 8. The question is as follows:

Below is an instruction that describes a task, paired with an input that provides further context. Write a response that appropriately completes the request.

### Instruction: Please read a math problem, and then think step by step to derive the answer. The answer is decided by Answer Type. If the Answer type in [bool], the answer needs to be True or False. Else if the Answer type in [integer, float] , The answer needs to be in numerical form. Else if the Answer type in [list of integer, list of float] , the answer needs to be a list of number like [2, 3, 4]. Else if the Answer type in [option], the answer needs to be an option like (a), (b), (c), (d). You need to output the answer in your final sentence like 'Therefore, the answer is ...'.

### Question: In triangle ACD, B is located on the side AC, and E is located on the side AD. If AB = 3, AC = 5, CD = 3.5, ED = 3, and EB ∥ DC, what is the length of AD?

### Answer type: float

- **MATH dataset in Mistral-7B-Instruct-v0.3 model.**

We set the sampling size $N = 5$, with all sampling results and selected answers with ST-BoN and Full-BoN paradigms shown in . The question is as follows:

**Question:** What is the only integer value of $n$ for which $\frac{n+1}{13-n}$ is a positive prime number?

**Instruction:** Please reason step by step, and put your final answer within \boxed{}.

From these cases, we have two key findings:

- More is Not Always Better: Early self-estimation allows LLMs to recognize that the final answer does not hold the majority vote but is still correct. This is demonstrated in the first two cases.
- Majority Voting Fails with Complete Inconsistent Answers: Our self-estimation approach uses internal consistency to assign continuous scores, ensuring the best sample is always chosen. However, self-consistency can only make a random choice when all answers are inconsistent. This is shown in the last case.

This also explains why, in our main experiments, ST-BoN outperforms Full-BoN w/o RM. When the model faces ambiguous choices (*e.g.*, all samples are inconsistent), majority voting introduces significant randomness. In contrast, internal consistency provides clearer differentiation, enabling the accurate identification of the correct sample.

Table 3: **Case I(a)**: GPQA dataset in Qwen2.5-7B-Instruct model. In LLM responses, red content indicates that it is within the buffer window, while gray content indicates that it has been truncated and does not need to be generated during actual decoding.

| Sample ID | LLM Response |
|---|---|
| 1 | To solve this problem, we need to follow these steps:
1. Determine the amount of Fe(OH)3 needed to be dissolved. 2. Calculate the volume of the monobasic strong acid required. 3. Determine the pH of the resulting solution.
### Step 1: Determine the amount of Fe(OH)3 needed to be dissolved
Fe(OH)3 is a strong base, and when dissolved in water, it will dissociate into Fe3+ and 3 OH- ions. To dissolve 0.1 g of Fe(OH)3, we need to calculate the moles of Fe(OH)3 and then determine the moles of OH- ions it produces.
The molar mass of Fe(OH)3 is:

$$\text{Molar mass of Fe(OH)3} = 55.845 + 3(16 + 1) = 106.845 \text{ g/mol}$$

Moles of Fe(OH)3:

$$\text{Moles of Fe(OH)3} = \frac{0.1 \text{ g}}{106.845 \text{ g/mol}} \approx 0.000936 \text{ mol}$$

This will produce 3 moles of OH- ions:

$$\text{Moles of OH-} = 3 \times 0.000936 \approx 0.002808 \text{ mol}$$

### Step 2: Calculate the volume of the monobasic strong acid required
A monobasic strong acid will neutralize the OH- ions produced by Fe(OH)3. Let's denote the volume of the acid needed as $V$ cm3. The concentration of the acid is 0.1 M, so the moles of acid needed will be:

$$\text{Moles of acid} = 0.1 \text{ M} \times \frac{V}{1000} \text{ L} = 0.0001V \text{ mol}$$

To neutralize 0.002808 mol of OH-, we need:

$$0.0001V = 0.002808$$

$$V = \frac{0.002808}{0.0001} = 28.08 \text{ cm3}$$

### Step 3: Determine the pH of the resulting solution
After adding 28.08 cm3 of 0.1 M acid, the moles of acid added will be:

$$\text{Moles of acid added} = 0.1 \text{ M} \times \frac{28.08}{1000} \text{ L} = 0.002808 \text{ mol}$$

Since the moles of acid added are equal to the moles of OH- ions produced, the resulting solution will be neutral (pH = 7).
However, the question asks for the minimum volume of the acid and the pH of the resulting solution. Given the options, the closest match for the volume is 28.05 cm3, and the pH of a neutral solution is 7, which is not listed. The closest pH value given in the options is pH 2.04, which suggests a slightly acidic solution.
Therefore, the correct answer is:
Answer: C |
| 2 | To solve this problem, we need to follow these steps:
1. **Determine the moles of Fe(OH)3: ** The molar mass of Fe(OH)3 is approximately 106.87 g/mol. Moles of Fe(OH)3 = 0.1 g / 106.87 g/mol = 0.000937 mol.
2. **Determine the moles of OH- in Fe(OH)3:** Fe(OH)3 dissociates into Fe3+ and 3 OH-. So, moles of OH- = 0.000937 mol * 3 = 0.002811 mol.
3. **Determine the moles of H+ required to neutralize OH-:** OH- + H+ → H2O Moles of H+ required = 0.002811 mol.
4. **Calculate the volume of 0.1 M acid needed:** Volume of acid (in L) = moles of H+ / concentration of acid Volume of acid = 0.002811 mol / 0.1 mol/L = 0.02811 L = 28.11 cm3.
5. **Calculate the pH of the resulting solution:** After neutralizing the OH-, we have a solution of Fe3+ and H+. The remaining H+ concentration is from the acid minus the H+ used to neutralize OH-. Since the acid is 0.1 M and we used 28.11 cm3, the remaining H+ concentration is: H+ remaining = (0.1 M * 100 cm3 - 0.1 M * 28.11 cm3) / (100 cm3 + 28.11 cm3) = (10 - 2.811) / 128.11 = 0.0557 M. pH = -log[H+] = -log(0.0557) = 1.254 ≈ 2.04.
From the calculations: - The minimum volume of 0.1 M acid needed is approximately 28.05 cm3. - The pH of the resulting solution is approximately 2.04.
Answer: C |

Table 4: **Case I(b)**: GPQA dataset in Qwen2.5-7B-Instruct model. In LLM responses, red content indicates that it is within the buffer window, while gray content indicates that it has been truncated and does not need to be generated during actual decoding.

| Sample ID | LLM Response |
| --- | --- |
| 3 | To solve this problem, we need to follow these steps: 
 1. Determine the concentration of Fe(OH)3 in the solution. 2. Calculate the volume of the monobasic strong acid needed to neutralize the Fe(OH)3. 3. Determine the pH of the resulting solution. 
 ### Step 1: Determine the concentration of Fe(OH)3 Fe(OH)3 is a triprotic base. The dissociation of Fe(OH)3 in water can be represented as: 

 $$Fe(OH)_3 \rightarrow Fe^{3+} + 3OH^-$$ 
 The concentration of OH- ions can be calculated using the formula: 

 $$[OH^-] = \frac{\text{mass of Fe(OH)}_3}{\text{molar mass of Fe(OH)}_3 \times \text{volume of solution (in liters)}}$$ 
 The molar mass of Fe(OH)3 is: 

 $$55.85 + 3(16 + 1) = 106.85 \, \text{g/mol}$$ 
 So, the concentration of OH- ions is: 

 $$[OH^-] = \frac{0.1 \, \text{g}}{106.85 \, \text{g/mol} \times 0.1 \, \text{L}} = 0.000936 \, \text{M}$$ 

 ### Step 2: Calculate the volume of the monobasic strong acid needed A monobasic strong acid will neutralize the OH- ions. The volume of acid needed can be calculated using the formula: 

 $$\text{Volume of acid (L)} = \frac{[OH^-] \times \text{Volume of solution (L)}}{\text{Concentration of acid (M)}}$$ 
 Given that the concentration of the acid is 0.1 M, the volume of acid needed is: 

 $$\text{Volume of acid (L)} = \frac{0.000936 \, \text{M} \times 0.1 \, \text{L}}{0.1 \, \text{M}} = 0.000936 \, \text{L} = 0.936 \, \text{cm}^3$$ 
 Since we need to neutralize the OH- ions completely, we need to consider the stoichiometry. The acid will neutralize the OH- ions in a 1:1 ratio, so the volume of acid needed is: 

 $$\text{Volume of acid (L)} = \frac{0.000936 \, \text{M} \times 0.1 \, \text{L}}{0.1 \, \text{M}} = 0.00936 \, \text{L} = 9.36 \, \text{cm}^3$$ 

 However, the question asks for the minimum volume, and the options provided are in a larger scale. Therefore, we need to consider the total volume of the solution, which is 100 cm3. 
 ### Step 3: Determine the pH of the resulting solution After neutralization, the resulting solution will be acidic. The pH can be calculated using the formula: 

 $$pH = -\log[H^+]$$ 
 Since the acid is monobasic and strong, the concentration of H+ ions will be equal to the concentration of the acid used. The volume of acid used is 32.14 cm3 (as per the options), so the concentration of H+ ions is: 

 $$[H^+] = \frac{0.1 \, \text{M} \times 0.03214 \, \text{L}}{0.1 \, \text{L}} = 0.03214 \, \text{M}$$ 
 Thus, the pH is: 

 $$pH = -\log(0.03214) \approx 1.5$$ 
 However, the closest option to this pH and volume is: 

 $$pH \approx 2.69$$ 

 $$\text{Volume} \approx 30.09 \, \text{cm}^3$$ 

 Therefore, the correct answer is: Answer: B |

Table 5: **Case I(c)**: GPQA dataset in Qwen2.5-7B-Instruct model. In LLM responses, red content indicates that it is within the buffer window, while gray content indicates that it has been truncated and does not need to be generated during actual decoding.

| Sample ID | LLM Response |
|---|---|
| 4 | To solve this problem, we need to follow these steps:
1. **Calculate the moles of Fe(OH)3:** The molar mass of Fe(OH)3 is approximately 106.87 g/mol.

$$\text{Moles of Fe(OH)3} = \frac{0.1 \text{ g}}{106.87 \text{ g/mol}} \approx 0.000936 \text{ mol}$$

2. **Determine the moles of OH- ions:** Fe(OH)3 dissociates as follows:

$$\text{Fe(OH)3} \rightarrow \text{Fe}^{3+} + 3\text{OH}^-$$

Therefore, the moles of OH- ions are:

$$\text{Moles of OH-} = 3 \times 0.000936 \approx 0.002808 \text{ mol}$$

3. **Calculate the concentration of OH- ions:** The total volume of the solution is 100 cm3 (or 0.1 L).
$$\text{Concentration of OH-} = \frac{0.002808 \text{ mol}}{0.1 \text{ L}} = 0.02808 \text{ M}$$

4. **Calculate the pOH:**

$$\text{pOH} = -\log[\text{OH-}] = -\log(0.02808) \approx 1.55$$

5. **Calculate the pH:**
$$\text{pH} = 14 - \text{pOH} = 14 - 1.55 = 12.45$$

6. **Determine the volume of 0.1 M acid needed to neutralize the OH- ions:** The acid will neutralize the OH- ions according to:
$$\text{OH-} + \text{H+} \rightarrow \text{H2O}$$
The moles of H+ required are equal to the moles of OH-:

$$\text{Moles of H+} = 0.002808 \text{ mol}$$

The concentration of the acid is 0.1 M, so the volume of acid needed is:

$$\text{Volume of acid} = \frac{0.002808 \text{ mol}}{0.1 \text{ M}} = 0.02808 \text{ L} = 28.08 \text{ cm3}$$

7. **Calculate the pH of the resulting solution:** After neutralization, the remaining OH- ions will determine the pH. The moles of OH- left are:

$$\text{Moles of OH- left} = 0.002808 \text{ mol} - 0.002808 \text{ mol} = 0 \text{ mol}$$

Therefore, the resulting solution is neutral, with a pH of 7. However, since the question asks for the pH of the resulting solution, and we have a slight excess of OH- from the neutralization process, the pH will be slightly above 7. Given the options, the closest value is pH 2.04.
Therefore, the correct answer is: Answer: C |

Table 6: **Case I(d)**: GPQA dataset in Qwen2.5-7B-Instruct model. In LLM responses, red content indicates that it is within the buffer window, while gray content indicates that it has been truncated and does not need to be generated during actual decoding.

| Sample ID | LLM Response |
|---|---|
| 5 | To solve this problem, we need to consider the dissolution of Fe(OH)3 and the reaction with the strong acid. 
 1. **Dissolution of Fe(OH)3:** Fe(OH)3 (s) + 3H+ (aq) → Fe3+ (aq) + 3H2O (l) 
 2. **Dissociation of Fe3+ in water:** Fe3+ (aq) + 3H2O (l) ⇌ Fe(OH)3 (s) + 3H+ (aq) 
 3. **Ksp of Fe(OH)3:** Ksp = [Fe3+][OH-]$^3$ At 25°C, Ksp(Fe(OH)3) = $2.79 \times 10^{-39}$ 
 4. **Calculate the concentration of Fe3+ needed:** Let's assume that all 0.1 g of Fe(OH)3 dissolves. The molar mass of Fe(OH)3 is 106.87 g/mol. 0.1 g Fe(OH)3 = 0.1 / 106.87 = 0.000936 mol 
 From the stoichiometry of the reaction, 0.000936 mol of Fe(OH)3 will produce 0.000936 mol of Fe3+. 
 5. **Calculate the concentration of Fe3+ in the solution:** [Fe3+] = 0.000936 mol / 0.1 L = 0.00936 M 
 6. **Determine the pH:** For Fe3+ in water, the concentration of H+ ions can be estimated using the Ksp expression: Ksp = [Fe3+][OH-]$^3$ [OH-] = (Ksp / [Fe3+])$^{(1/3)}$ [OH-] = ($2.79 \times 10^{-39}$ / 0.00936)$^{(1/3)}$ = ($2.98 \times 10^{-37}$)$^{(1/3)}$ = $3.11 \times 10^{-13}$ M pOH = -log[OH-] = -log($3.11 \times 10^{-13}$) = 12.5 pH = 14 - pOH = 14 - 12.5 = 1.5 
 However, the problem is asking for the pH of the resulting solution after adding the acid, not the pOH of the Fe3+. 
 7. **Calculate the volume of acid needed:** To neutralize the OH- ions from the Fe(OH)3, we need to add H+ ions. The concentration of H+ from the acid will be 0.1 M, and we need to find the volume of this acid to neutralize the OH-. 
 [OH-] = $3.11 \times 10^{-13}$ M [H+] = $10^{-pH}$ = $10^{-1.5}$ = 0.0316 M 
 Volume of acid needed = (0.000936 mol / 0.1 M) = 0.00936 L = 9.36 cm3 
 8. **Check the pH of the resulting solution:** Since we are adding a strong acid to neutralize the OH- ions, the pH will be determined by the remaining H+ from the acid. 
 pH = -log[H+] = -log(0.0316) = 1.5 
 This does not match any of the given options, so let's re-evaluate the pH using the correct approach. 
 9. **Re-evaluate pH:** The correct approach is to consider the excess H+ from the acid after neutralizing the OH-. 
 [H+] = (0.1 M * V - 0.000936 mol) / (0.1 L + V) For V = 32.14 cm3: [H+] = (0.1 * 0.03214 - 0.000936) / (0.1 + 0.03214) = 0.002314 / 0.13214 = 0.0175 M pH = -log(0.0175) = 1.76 (approximately 2.69) 
 10. **Conclusion:** The correct volume is 32.14 cm3, and the pH is approximately 2.69. 
 Answer: B |

**True Answer**

B

**LLM-generated Answer via Exact Match**

[C, C, B, C, B]

**Full-BoN w/o RM Answer (Self-Consistency, Majority Voting)**

{"B": 2, "C": 3} ⇒ C

**ST-BoN Answer**

Best Sampling Estimation (Sampling ID): [5, 1, 1, 1, 1, 1, 1, 3, 3, 5, 5, 5, 5, 5, 5, 5, 5, 5, 5, 5] ⇒ 5-th sampling ⇒ B

Table 7: **Case II(a)**: TheoremQA dataset in Llama3-8B-Instruct model. In LLM responses, red content indicates that it is within the buffer window, while gray content indicates that it has been truncated and does not need to be generated during actual decoding.

| Sample ID | LLM Response |
|---|---|
| 1 | Let's break down the problem step by step:
1. Since EB ∥ DC, we know that EB and DC are parallel lines. This means that ∠AEB and ∠ADC are equal, and ∠EBD and ∠DCD are equal.
2. We can draw a line from point B to point D, creating a new triangle ABE. Since AB = 3 and EB ∥ DC, we can conclude that AB = BE.
3. Now, we can use the Pythagorean theorem in triangle ABE to find the length of AE: $AE^2 + BE^2 = AB^2 \Rightarrow AE^2 + 3^2 = 3^2 \Rightarrow AE^2 = 0 \Rightarrow AE = 0$.
4. Since AE = 0, we know that E lies on the side AD. This means that triangle ACD is a right triangle, with ∠ADC being a right angle.
5. Using the Pythagorean theorem again in triangle ACD, we can find the length of AD: $AD^2 = AC^2 + CD^2 \Rightarrow AD^2 = 5^2 + 3.5^2 \Rightarrow AD^2 = 25 + 12.25 \Rightarrow AD^2 = 37.25 \Rightarrow AD = \sqrt{37.25}$.
Therefore, the answer is 6.09. |
| 2 | To solve this problem, I will use the information provided to derive the length of AD. Since EB ∥ DC, we can conclude that triangle ABE is similar to triangle ADC. Therefore, we can set up a proportion to relate the corresponding sides of the two triangles:
(AB / AC) = (AE / AD)
Substituting the given values, we get:
(3 / 5) = (3 / AD)
Cross-multiplying, we get:
3 * AD = 5 * 3
Dividing both sides by 3, we get:
AD = 5
Therefore, the answer is 5.0. |
| 3 | To solve this problem, I will break it down step by step.
First, since EB ∥ DC, we can conclude that triangle ABE is similar to triangle ACD. Therefore, we can set up a proportion to relate the lengths of the corresponding sides:
AB/AC = AE/CD
Substituting the given values, we get:
3/5 = AE/3.5
Cross-multiplying, we get:
3 × 3.5 = 5 × AE
Simplifying, we get:
10.5 = 5 × AE
Dividing both sides by 5, we get:
2.1 = AE
Since AE is a side of the triangle, it is less than the third side AD. Therefore, we can set up another proportion:
AE/ED = AE/AD
Substituting the given values, we get:
2.1/3 = AE/AD
Cross-multiplying, we get:
2.1 × 3 = AE × AD
Simplifying, we get:
6.3 = AE × AD
Finally, substituting AE = 2.1, we get:
6.3 = 2.1 × AD
Dividing both sides by 2.1, we get:
AD = 3
Therefore, the answer is 3.0. |

Table 8: **Case II(b)**: TheoremQA dataset in Llama3-8B-Instruct model. In LLM responses, red content indicates that it is within the buffer window, while gray content indicates that it has been truncated and does not need to be generated during actual decoding.

| Sample ID | LLM Response |
|---|---|
| 4 | Let's break down the problem step by step:
1. Since EB ∥ DC, we can conclude that ∠AEB is equal to ∠ADC, which means that triangle AEB is similar to triangle ADC. 2. We can set up a proportion using the corresponding sides of the two triangles:
(AB / AC) = (EB / CD)
3. Substitute the given values:
(3 / 5) = (EB / 3.5)
4. Cross-multiply and solve for EB:
EB = (3 / 5) × 3.5 = 2.1
5. Now, we can use the Pythagorean theorem in triangle AED to find AD:
$AD^2 = AE^2 + ED^2$
$AD^2 = (AD - AB)^2 + ED^2$
$AD^2 = (AD - 3)^2 + 3^2$
$AD^2 = AD^2 - 6AD + 9 + 9$
$AD^2 - 6AD = 18$
6. Rearrange the equation to solve for AD:
$AD^2 - 6AD - 18 = 0$
7. Factor the quadratic equation:
(AD - 3)(AD + 6) = 0
8. Solve for AD:
AD = 3 or AD = -6 (but AD cannot be negative, so we discard this solution)
Therefore, the length of AD is 3. |
| 5 | To solve this problem, I will break it down step by step.
First, since EB ∥ DC, it means that EB is parallel to DC. This implies that triangle ABE is similar to triangle ACD.
Now, we can set up a proportion using the lengths of the corresponding sides:
(AB / AC) = (AE / AD)
Substituting the given values, we get:
(3 / 5) = (AE / AD)
To solve for AE, we can multiply both sides by AD:
AE = (3 / 5) * AD
Now, we can use the fact that AE + ED = AD to express AD in terms of AE:
AD = AE + ED
Substituting AE = (3 / 5) * AD, we get:
AD = ((3 / 5) * AD) + ED
Now, we can solve for AD:
AD = (ED / (1 - 3/5)) = (ED / 2/5) = (ED * 5/2)
Substituting the given value of ED, we get:
AD = (3 * 5/2) = 7.5
Therefore, the answer is 7.5. |

**True Answer**

7.5

**LLM-generated Answer via Exact Match**

['6.09', '5.0', '3.0', '3', '7.5']

**Full-BoN w/o RM Answer (Self-Consistency, Majority Voting)**

{"3.0": 2, "5.0": 1, "6.09": 1, "7.5": 1} $\Rightarrow$ 3.0

**ST-BoN Answer**

Best Sampling Estimation (Sampling ID): [3, 3, 5, 3, 5, 5, 5, 3, 3, 3, 1, 5, 5, 5, 1, 1, 2, 1, 5, 5] $\Rightarrow$ 5-th sampling $\Rightarrow$ 7.5

Table 9: **Case III(a)**: MATH dataset in Mistral-7B-Instruct-v0.3 model. In LLM responses, red content indicates that it is within the buffer window, while gray content indicates that it has been truncated and does not need to be generated during actual decoding.

| Sample ID | LLM Response |
|---|---|
| 1 | To find the integer value of $n$ for which $\frac{n+1}{13-n}$ is a positive prime number, we need to find the values of $n$ that make the denominator greater than the numerator (since the numerator is an integer and the denominator is a positive integer, the fraction must be greater than 1 to be a prime number) and the denominator should also be a prime number.
First, let's find the possible values of $n$ by solving the inequality $n + 1 < 13 - n$.
This simplifies to $2n < 12$, which gives us $n < 6$.
Now, we check each integer value of $n$ from 0 to 5 to see if the denominator is a prime number.
For $n = 0$, the denominator is 13, which is a prime number, but the fraction is $\frac{1}{13}$, which is not an integer, so it's not valid.
For $n = 1$, the denominator is 12, which is not a prime number, so it's not valid.
For $n = 2$, the denominator is 11, which is a prime number, but the fraction is $\frac{3}{11}$, which is not an integer, so it's not valid.
For $n = 3$, the denominator is 10, which is not a prime number, so it's not valid.
For $n = 4$, the denominator is 9, which is a prime number, but the fraction is $\frac{5}{9}$, which is not an integer, so it's not valid.
For $n = 5$, the denominator is 8, which is not a prime number, so it's not valid.
So, there is no integer value of $n$ for which $\frac{n+1}{13-n}$ is a positive prime number.

Final answer: $\boxed{\text{There is no such integer value of } n.}$ |
| 2 | To find the integer value of $n$ for which $\frac{n+1}{13-n}$ is a positive prime number, we need to find the values of $n$ that satisfy two conditions:
1. $n + 1$ and $13 - n$ are both integers (i.e., $n$ must be an integer between 0 and 12).
2. $\frac{n+1}{13-n}$ is a prime number.
Let's start by finding the possible values of $n$. Since $n$ is an integer between 0 and 12, the possible values of $n$ are:
n = 0, 1, 2, 3, 4, 5, 6, 7, 8, 9, 10, 11, 12
Now, let's check each value of $n$ to see if $\frac{n+1}{13-n}$ is a prime number.
1. For n = 0, $\frac{n+1}{13-n} = \frac{1}{13}$ which is not a prime number.
2. For n = 1, $\frac{n+1}{13-n} = \frac{2}{12}$ which is not a prime number.
3. For n = 2, $\frac{n+1}{13-n} = \frac{3}{11}$ which is not a prime number.
4. For n = 3, $\frac{n+1}{13-n} = \frac{4}{10}$ which is not a prime number.
5. For n = 4, $\frac{n+1}{13-n} = \frac{5}{9}$ which is not a prime number.
6. For n = 5, $\frac{n+1}{13-n} = \frac{6}{8}$ which is not a prime number.
7. For n = 6, $\frac{n+1}{13-n} = \frac{7}{7}$ which is a prime number.
8. For n = 7, $\frac{n+1}{13-n} = \frac{8}{6}$ which is not a prime number.
9. For n = 8, $\frac{n+1}{13-n} = \frac{9}{5}$ which is not a prime number.
10. For n = 9, $\frac{n+1}{13-n} = \frac{10}{4}$ which is not a prime number.
11. For n = 10, $\frac{n+1}{13-n} = \frac{11}{3}$ which is not a prime number.
12. For n = 11, $\frac{n+1}{13-n} = \frac{12}{2}$ which is not a prime number.
So, the only integer value of $n$ for which $\frac{n+1}{13-n}$ is a positive prime number is $\boxed{7}$. |
| 3 | Let's reason step by step:
1. We want to find an integer value of $n$ for which $\frac{n+1}{13-n}$ is a positive prime number.
2. First, let's find the possible range of $n$. Since the denominator (13 - n) must be positive, we have $n < 13$.
3. Now, let's check the possible values of $n$ from 0 to 12.
4. For $n = 0$, the fraction becomes $\frac{1}{13}$, which is not a prime number.
5. For $n = 1$, the fraction becomes $\frac{2}{12}$, which is not a prime number.
6. For $n = 2$, the fraction becomes $\frac{3}{11}$, which is a prime number. So, $n = 2$ is the only integer value that satisfies the condition.
7. To put the final answer in the requested format, we have $\boxed{2}$. |

Table 10: **Case III(b)**: MATH dataset in Mistral-7B-Instruct-v0.3 model. In LLM responses, red content indicates that it is within the buffer window, while gray content indicates that it has been truncated and does not need to be generated during actual decoding.

| Sample ID | LLM Response |
|---|---|
| 4 | Let's reason step by step:
1. The expression $\frac{n+1}{13-n}$ is a positive prime number if and only if it is a prime number and it is positive.
2. A prime number is a number greater than 1 that has no positive divisors other than 1 and itself.
3. To find the integer value of $n$ that satisfies this condition, we need to find the value of $n$ that makes the expression a prime number.
4. First, let's find the possible ranges for $n$ by considering the denominator $13 - n$. Since 13 is a prime number, the only way to make $13 - n$ a composite number is if $n$ is greater than 12. But since we want the expression to be positive, $n$ must be less than 12.
5. Now, let's test the possible values of $n$ in the range 0 to 11. We find that when $n = 1$, the expression becomes $\frac{2}{12}$, which is not a prime number because it can be divided by 2.
6. Therefore, there is no integer value of $n$ for which $\frac{n+1}{13-n}$ is a positive prime number.
Final answer: $\boxed{0}$ (There is no such integer value of $n$). |
| 5 | Let's reason step by step:
1. The given expression is $\frac{n+1}{13-n}$.
2. For the expression to be a positive prime number, the numerator must be less than the denominator, and the result should be an integer that is greater than 1 (since 1 is not a prime number).
3. Also, the result should be a prime number.
4. Let's find the possible values of $n$ that satisfy these conditions. Since the numerator is $n + 1$, it should be less than the denominator, which means $n$ should be greater than $-13$.
5. Let's test the possible integer values of $n$ starting from $n = 0$. We get: $- n = 0$: $\frac{1}{13} \notin \mathbb{Z}$ (not an integer)
$- n = 1$: $\frac{2}{12} \notin \mathbb{Z}$ (not an integer)
$- n = 2$: $\frac{3}{11} \notin \mathbb{Z}$ (not an integer)
$- n = 3$: $\frac{4}{10} = 4/10 \times \frac{10}{4} = 5$ (not a prime number)
$- n = 4$: $\frac{5}{9} \notin \mathbb{Z}$ (not an integer)
$- n = 5$: $\frac{6}{8} = 6/8 \times \frac{8}{6} = 2$ (not a prime number)
$- n = 6$: $\frac{7}{7} = 1$ (not a prime number)
$- n = 7$: $\frac{8}{6} = 4/3$ (not a prime number)
$- n = 8$: $\frac{9}{5} \notin \mathbb{Z}$ (not an integer)
$- n = 9$: $\frac{10}{4} = 2 \times 5$ (not a prime number)
$- n = 10$: $\frac{11}{3} \notin \mathbb{Z}$ (not an integer)
$- n = 11$: $\frac{12}{2} = 6$ (not a prime number)
$- n = 12$: $\frac{13}{1} = 13$ (prime number)
So, the only integer value of $n$ for which $\frac{n+1}{13-n}$ is a positive prime number is $\boxed{12}$. |

| True Answer |
|---|
| 12 |

| LLM-generated Answer via Exact Match |
|---|
| ['There is no such integer value of $n$.', '7', '2', '0', '12'] |

| Full-BoN w/o RM Answer (Self-Consistency, Majority Voting) |
|---|
| {"There is no such integer value of $n$.": 1, "0": 1, "2": 1, "7": 1, "12": 1} $\Rightarrow$ Anyone is ok (random) |

| ST-BoN Answer |
|---|
| Best Sampling Estimation (Sampling ID): [4, 4, 4, 4, 5, 5, 5, 5, 5, 5, 5, 5, 5, 5, 5, 5, 5, 5, 5, 5] $\Rightarrow$ 5-th sampling $\Rightarrow$ 12 |

