# OpenReview forum: "Sampling-Efficient Test-Time Scaling: Self-Estimating the Best-of-N Sampling in Early Decoding"
_NeurIPS.cc/2025/Conference — NeurIPS 2025 spotlight_

### Official Review · Reviewer_GGmq · 2025-06-28

**Clarity:** 4
**Significance:** 4
**Originality:** 4
**Rating:** 5
**Confidence:** 4

**Summary:**

This work is motivated by a new insight found by the authors: early estimated consistency is correlated with final consistency. Thus, a sample that is closer to others in early decoding is more likely to reach the correct answer when decoding ends. Following this insight, the authors propose to early stop inconsistent sampling at an early stage using consistency measured by hidden states, thereby greatly reducing the GPU memory and latency required for Best-of-N sampling. The results are promising: much better than naive BoN and BoN with reward, even though no additional reward model is needed.

**Questions:**

1. From Figure 5, it looks like ST-BoN can achieve better accuracy with 1× computational cost compared to greedy. In my understanding, ST-BoN is using best-of-N sampling and should always require more computation than greedy, so how can their computation be the same?

2. Experiments in this paper are conducted on non-deep thinking models and tasks with short thinking lengths. For deep reasoning models such as Qwen3, and benchmarks that require deep thinking in order to get the correct answer, such as AIME, they will produce or need very long reasoning chains. In that situation, the early consistency prediction may not be accurate, as the model may reflect and self-correct at a later stage, although they may look the same at the beginning. Can your method still apply to this situation?

**Ethical Concerns:**

["NO or VERY MINOR ethics concerns only"]

**Final Justification:**

The rebuttal has clarified my questions and provided additional experimental results, especially those comparing with Full-BoN on long reasoning tasks. It is exciting to see such a great accuracy-efficiency trade-off for the proposed method. I will keep my accept rating as my final justification, as this is solid work.

**Limitations:**

See question 2 in Questions Section.

**Paper Formatting Concerns:**

No concern.

**Quality:**

4

**Strengths And Weaknesses:**

**Quality**: This work is well supported by both clear theoretical analysis and positive experimental results, showing very promising results on various tasks.

**Clarity**: The structure of the paper is very clear, with core information highlighted, making it easy to follow and understand. The authors start with a hypothesis, then provide a theoretical analysis to support the claim, motivating their method from the claim and finally presenting extensive experiments to support the results. It is very fluent.

**Significance**: The overhead from BoN is a critical issue, and the proposed method does not rely on any additional model or training to successfully reduce the overhead, which is significant and insightful and can motivate more work exploring consistency during sampling.

**Originality**: Using inconsistency to guide early termination is novel, and this idea can bring great performance gains with less computation and without the need for any additional reward models, making it a simple but effective work.

---

> ### Author Rebuttal · Authors · 2025-07-27
>
> Thank you very much for recognizing the **Quality**, **Clarity**, **Significance**, and **Originality** of our work. Regarding your two questions (Q1-Q2), we address them one by one as follows:
>
> ---
>
>
> > Q1: Regarding the 1x Cost of ST-BoN.
>
>
> It is evident that the computational cost of ST-BoN cannot be identical to that of greedy decoding. As you noted, it requires at least a brief $N$ sampling.
>
> We would like to clarify this potential misunderstanding. Due to the limited space in Figure 5, the computational costs for $N = 3$ and $N = 5$ **appear visually close to the 1x baseline, but actually not the same**. The precise computational costs for the four objective tasks are as follows:
>
> ||$N=3$|$N=5$|
> |-|-|-|
> |MATH|1.11x|1.22x|
> |TheoremQA|1.21x|1.32x|
> |GPQA|1.20x|1.29x|
> |MMLU|1.35x|1.48x|
>
> We will include these numerical values in the revised appendix to enhance clarity and transparency.
>
> ---
>
> > Q2: Regarding the Applicability to Reasoning Models
>
> Thanks for your suggestion regarding the broader applicability of ST-BoN. To evaluate the applicability of ST-BoN on long-CoT reasoning models, we report results on **Qwen3-8B-Thinking** and **Deepseek-R1-Distill-Qwen-7B**, using the **AIME24+25** dataset. Both the models and dataset are representative of long-CoT paradigm.
> (*Due to computational resource constraints, Full-BoN was only evaluated up to $ N \leq 40$.*)
>
> *Qwen3-8B-Thinking (Accuracy / Cost):*
>
> |                 | $N=10$         | $N=20$     | $N=40$         | $N=80$    | $N=160$       |
> | --------------- | ---------------- | ------------ | ---------------- | ----------- | --------------- |
> | Full-BoN w/o RM | **68.3 / 8.1×**  | 68.3 / 11.5× | 70.0 / 19.6×     | —           | —               |
> | Full-BoN w/ RM  | **66.7 / 10.2×** | 70.0 / 14.6× | **73.3 / 23.2×** | —           | —               |
> | ST-BoN (ours)   | 65.0 / 1.2×      | 68.3 / 1.4×  | 68.3 / 1.7×      | 71.7 / 2.2× | **73.3 / 3.2×** |
>
> *Deepseek-R1-Distill-Qwen-7B (Accuracy / Cost):*
>
> |                 | $N=10$         | $N=20$     | $N=40$         | $N=80$    | $N=160$       |
> | --------------- | --------------- | ------------ | ---------------- | ----------- | --------------- |
> | Full-BoN w/o RM | **70.0 / 7.2×** | 75.0 / 10.3× | **75.0 / 17.4×** | —           | —               |
> | Full-BoN w/ RM  | **71.7 / 9.8×** | 73.3 / 13.7× | **75.0 / 21.8×** | —           | —               |
> | ST-BoN (ours)   | 68.3 / 1.3×     | 71.7 / 1.5×  | 73.3 / 1.8×      | 73.3 / 2.5× | **75.0 / 3.4×** |
>
> *Key insights:*
>
> * *First*, due to the much longer generation lengths of reasoning models (often >10k tokens) compared to non-reasoning tasks (often <1k), the memory cost of KV cache becomes dominant. This significantly inflates the overall cost of Full-BoN. In contrast, ST-BoN performs early truncation and retains only a single path up to ~10k tokens, keeping its cost only marginally higher than that of greedy decoding.
>
> * *Second*, ST-BoN still exhibits a markedly better cost–performance trade-off. For instance, on Qwen3-8B-Thinking, to reach 73.3 accuracy, **ST-BoN requires only 13.8% (3.2/23.2) of the cost of Full-BoN w/ RM**. Notably, even when ST-BoN uses $N=160$, Full-BoN at $N=10$ remains more expensive while underperforming.
>
> **These results provide strong evidence for the scalability and efficiency of ST-BoN in the context of long-CoT reasoning models.**
>
> > *Further Insight:* To explain this phenomenon, we find support from a recent study [1], which shows that the accuracy of the *first reasoning step* often predicts final success, even when later decoding involves rollback or reflection. This further supports the effectiveness behind ST-BoN’s early truncation design in any LLM paradigm.
> [1] *Lost at the Beginning of Reasoning*. [https://arxiv.org/abs/2506.22058](https://arxiv.org/abs/2506.22058)
>
> ---
>
> We hope these results satisfactorily address your concerns and underscore the broader significance of our contributions. We look forward to your positive feedback, thanks again for your constructive comments.

---

> ### Comment · Reviewer_GGmq · 2025-07-31
>
> Thank you for clarifying my questions and providing additional experimental results, especially those for comparing with Full-BoN on long reasoning tasks.. It is exciting to see such a great accuracy-efficiency trade-off for your proposed method. I will keep my accept rating as this is a solid work.

---

> > ### Author Response · Authors · 2025-08-01
> >
> > Thanks for your positive comments on our work! We truly appreciate your support.

---

### Official Review · Reviewer_9oj8 · 2025-06-29

**Clarity:** 3
**Significance:** 3
**Originality:** 3
**Rating:** 5
**Confidence:** 4

**Summary:**

In light of recent research findings on LLM's self-consistency, and that their early hidden states already encode substantial information about the full continuation, the authors proposed Self-Truncation Best-of-N decoding(ST-BoN) that addresses the compute and memory limitations of conventional Best-of-N decoding. Specifically, ST-BoN conducts final self-consistency estimation based truncation by leveraging latent information starting at the early decoding steps where the candidate beams fully diverged without the reliance on textual heuristics or reward models.

**Questions:**

- Will the code be released?

Suggestions:
- It might be better to rephrase early decoding as early decoding steps at some places in the paper for better clarity.
- The title may need to be revised to clearly and comprehensively reflect the paper’s core contribution and align with its content.

**Ethical Concerns:**

["NO or VERY MINOR ethics concerns only"]

**Final Justification:**

The author’s response slightly mitigated my concern about the weakness, though the comprehensive results and analysis, as well as the stronger baselines mentioned by other reviewers, should be included in the final version of the paper. I believe my original rating fully reflected the value of the work, and I would like to keep it as is.

**Limitations:**

yes

**Quality:**

4

**Strengths And Weaknesses:**

Strengths:
- The proposed method is well-motivated with recent research findings and theoretical proof.
- The effectiveness and efficiency of ST-BoN have been validated by extensive experiments and ablation studies, and ST-BoN is competitive on both objective and subjective tasks.
- The relation of buffer window length with respect to earliest estimation time/length is well studied, and it suggests that the approach has a generalizable proportional constant.

Weaknesses:
- The sensitivity of the proposed method with respect to the target generation length is understudied.

---

> ### Author Rebuttal · Authors · 2025-07-27
>
> Thank you very much for recognizing our work. We especially appreciate your acknowledgment of its well-motivated and theoretically grounded design, extensive experiments/ablations, and strong generalization. Regarding the weaknesses and questions you raised, we address them one by one as follows.
>
>
> ---
>
> > W: The sensitivity of ST-BoN with respect to the target generation length is understudied.
>
> Thank you for pointing this out. To sensitivity of ST-BoN with respect to the target generation length, we choose some long-CoT reasoning models to demonstrate this issue. These models exhibit a "slow-thinking" characteristic, often producing outputs that are significantly longer than those of "fast-thinking" non-reasoning models. Specifically, we report results on **Qwen3-8B-Thinking** and **Deepseek-R1-Distill-Qwen-7B**, using the **AIME24+25** dataset.
> (*Due to computational resource constraints, Full-BoN was only evaluated up to $N \leq 40$.*)
>
> *Qwen3-8B-Thinking (Accuracy / Cost):*
>
> |                 | $N=10$         | $N=20$     | $N=40$         | $N=80$    | $N=160$       |
> | --------------- | ---------------- | ------------ | ---------------- | ----------- | --------------- |
> | Full-BoN w/o RM | **68.3 / 8.1×**  | 68.3 / 11.5× | 70.0 / 19.6×     | —           | —               |
> | Full-BoN w/ RM  | **66.7 / 10.2×** | 70.0 / 14.6× | **73.3 / 23.2×** | —           | —               |
> | ST-BoN (ours)   | 65.0 / 1.2×      | 68.3 / 1.4×  | 68.3 / 1.7×      | 71.7 / 2.2× | **73.3 / 3.2×** |
>
> *Deepseek-R1-Distill-Qwen-7B (Accuracy / Cost):*
>
> |                 | $N=10$         | $N=20$     | $N=40$         | $N=80$    | $N=160$       |
> | --------------- | --------------- | ------------ | ---------------- | ----------- | --------------- |
> | Full-BoN w/o RM | **70.0 / 7.2×** | 75.0 / 10.3× | **75.0 / 17.4×** | —           | —               |
> | Full-BoN w/ RM  | **71.7 / 9.8×** | 73.3 / 13.7× | **75.0 / 21.8×** | —           | —               |
> | ST-BoN (ours)   | 68.3 / 1.3×     | 71.7 / 1.5×  | 73.3 / 1.8×      | 73.3 / 2.5× | **75.0 / 3.4×** |
>
> *Key insights:*
>
> * *First*, due to the much longer generation lengths of reasoning models (often >10k tokens) compared to non-reasoning tasks (often <1k), the memory cost of KV cache becomes dominant. This significantly inflates the overall cost of Full-BoN. In contrast, ST-BoN performs early truncation and retains only a single path up to ~10k tokens, keeping its cost only marginally higher than that of greedy decoding.
>
> * *Second*, ST-BoN still exhibits a markedly better cost–performance trade-off. For instance, on Qwen3-8B-Thinking, to reach 73.3 accuracy, **ST-BoN requires only 13.8% (3.2/23.2) of the cost of Full-BoN w/ RM**. Notably, even when ST-BoN uses $N=160$, Full-BoN at $N=10$ remains more expensive while underperforming.
>
> **These results provide strong evidence that ST-BoN consistently achieves a favorable cost–performance trade-off across both short and long output scenarios, *reinforcing the robustness with respect to the target generation length of our ST-BoN.***
>
> ---
>
> > Q1-Q3
>
> * Code: Of course, we are preparing a clean version of the code, which will be released once the publication.
>
> * Writing: Thanks for your suggestions on improving our writing. We will give them full consideration and incorporate the revisions in the next version.
>
> ---
>
> We hope these results satisfactorily address your concerns and underscore the broader significance of our contributions. We look forward to your positive feedback, thanks again for your constructive comments.

---

### Official Review · Reviewer_6txQ · 2025-06-30

**Clarity:** 3
**Significance:** 1
**Originality:** 1
**Rating:** 4
**Confidence:** 5

**Summary:**

The authors present a decoding method (ST-BON) as an alternative to Best-of-N sampling, that forgoes the need to generate all N samples and eliminates the need of reward models. To do so, it measures early sample consistency to identify the most promising path and then prunes out suboptimal paths. With respect to the tradeoff of cost to performance, for ST-BON, reduces GPU memory usage, and inference latency, while preserving performance. Alternatively, with the same cost, it can improve accuracy of the model further.

**Questions:**

1. Baselines that includes partial traces and iterative refinement such as [1,2,5] were not compared against. How does your work compare to that?
2. Generative reward models [3,4] which reduce the flop usage are not compared against. How does your work compare to this?
3. For the objective tasks, have you compared the performance of your approach to a rule-based (oracle) reward model to evaluate the bias of the PRM. It is possible that the PRM isn't trained on on-policy samples leading to a large bias in the PRM estimate.
4. Subjective tasks evaluation done with a RM trained on QA preferences but evaluated on summarization seems to be a reward mispecification problem rather than a benefit of your approach. Could you evaluate how an RM trained on the summarization task would compare.
5. Have you done evaluation with Long-COT models such as DeepSeek-R1 or Qwen 3 for your approach. How do these models fair?

References
[1] Scaling LLM Test-Time Compute Optimally can be More Effective than Scaling Model Parameters
[2] ReST-MCTS: LLM Self-Training via Process Reward Guided Tree Search
[3] Generative Verifiers: Reward Modeling as Next-Token Prediction
[4] Generative Reward Models
[5] Adaptive Inference-Time Compute: LLMs Can Predict if They Can Do Better, Even Mid-Generation

**Ethical Concerns:**

["NO or VERY MINOR ethics concerns only"]

**Final Justification:**

The rebuttal experiments address my concerns. I thus raise it to a 4.

**Limitations:**

yes

**Quality:**

3

**Strengths And Weaknesses:**

Strengths
- Provide extensive empirical evaluation of their method

Weaknesses
- Baseline Concerns: Approaches that do partial pruning (e.g Beam Search[1], MCTS[2], or Adaptive Inference-Time Compute[5]) or learn a generative verifier [3,4] (and reuse the KV cache during inference) are not comprehensively compared against in this work.
- Limited Novelty: Metric is heavily based on a prior work (chain of embedding).

References
[1] Scaling LLM Test-Time Compute Optimally can be More Effective than Scaling Model Parameters
[2] ReST-MCTS: LLM Self-Training via Process Reward Guided Tree Search
[3] Generative Verifiers: Reward Modeling as Next-Token Prediction
[4] Generative Reward Models
[5] Adaptive Inference-Time Compute: LLMs Can Predict if They Can Do Better, Even Mid-Generation

---

> ### Author Rebuttal · Authors · 2025-07-29
>
> Thank you for your constructive feedback. Below, we address the weaknesses (W1–W2) and questions (Q1–Q5) you raised one by one:
>
> ---
>
> > W1: Baseline Concerns
>
> Thank you for pointing out these potential baselines. We present a Summary Table here and provide detailed explanations in responses to **Q1 and Q2**.
>
> # Summary Table
>
> Model: Qwen2.5-7B-Instruct
>
> * *MATH (Accuracy / Cost):*
>
> ||$N=5$|$N=10$|$N=20$|$N=30$|$N=60$|$N=80$|$N=120$|
> |-|-|-|-|-|-|-|-|
> |Full-BoN w/ RM|72.4 / 5.1x|73.3 / 6.8x|**74.2 / 10.2x**|75.0 / 13.2x|——|——|——|
> |Full-BoN w/ GRM [3,4]|72.8 / 5.8x|73.9 / 7.8x|**74.6 / 11.5x**|75.5 / 14.6x|——|——|——|
> |Beam Search [1]|70.7 / 3.7x|71.9 / 7.6x|73.5 / 14.4x|**74.2 / 22.6x**|——|——|——|
> |MCTS [2]|71.7 / 9.1x|**74.3 / 11.6x**|75.3 / 19.5x|76.0 / 28.1x|——|——|——|
> |Adaptive [5]|70.1 / 3.2x|71.5 / 5.5x|72.1 / 12.4x|73.3 / 18.7x|——|——|——|
> |ST-BoN|70.3 / 1.2x|71.1 / 1.3x|71.8 / 1.6x|72.8 / 1.9x|74.1 / 2.9x|**74.5 / 3.6x**|75.0 / 4.9x|
>
> * *GPQA (Accuracy / Cost):*
>
> ||$N=5$|$N=10$|$N=20$|$N=30$|$N=60$|$N=80$|$N=120$|
> |-|-|-|-|-|-|-|-|
> |Full-BoN w/ RM|33.8 / 4.9x|34.3 / 6.6x|35.3 / 10.0x|**35.7 / 13.1x**|——|——|——|
> |Full-BoN w/ GRM [3,4]|33.4 / 5.6x|34.7 / 7.7x|36.2 / 11.0x|**36.5 / 14.4x**|——|——|——|
> |Beam Search [1]|32.0 / 3.4x|33.2 / 7.2x|34.8 / 13.6x|**36.0 / 21.8x**|——|——|——|
> |MCTS [2]|33.1 / 7.8x|33.9 / 11.1x|**36.2 / 18.6x**|37.1 / 27.5x|——|——|——|
> |Adaptive [5]|32.5 / 3.6x|33.3 / 7.7x|33.7 / 13.4x|**35.4 / 17.1x**|——|——|——|
> |ST-BoN|32.3 / 1.3x|33.3 / 1.4x|33.5 / 1.7x|34.9 / 2.0x|36.1 / 3.1x|**36.4 / 3.7x**|36.9 / 4.8x|
>
> ---
>
> > W2: Limited Novelty: Metric is heavily based on a prior work (chain of embedding).
>
> ***<Please refer to Reviewer QemV’s "W3(1)" for a detailed response.>***
>
> Overall, the core idea of ST-BoN lies in the *self-truncating BoN framework*. The chain-of-embedding consistency metric is more about an empirical contribution. **The two parts are complementary and equally essential.**
>
> **Notably, both Reviewer QemV and Reviewer GGmq explicitly praised the novelty of ST-BoN.**
>
> ---
>
> > Q1: Beam Search[1], MCTS[2], or Adaptive Inference-Time Compute[5], were not compared.
>
> Our research aims to optimize the specific *best-of-N* paradigm. From the perspective of the search paradigm, **[1], [2], and our approach are orthogonal --- as illustrated in Figure 2 of [1], the three types of methods are fundamentally independent**:
>
> * *ST-BoN* (ours) represents a **randomized** global sampling search,
> * *Beam Search*[1] performs a local **heuristic** search,
> * *MCTS*[2] performs a global tree-based **heuristic** search driven by simulation and value backpropagation.
>
> Only *Adaptive*[5] directly targets *best-of-N* optimization.
>
> Despite this, we agree that comparing against these methods can provide valuable insights and strengthen connections with parallel lines of research. Since [1] and [2] do not involve random sampling, they do not have the specific hyperparameter $N$. For fair comparison, we follow [1] by setting beam width and MCTS expansion to $[\sqrt{N}]$. Results are shown in the summary table in W1:
>
> * **Cost:**
>
>   * *Beam Search* must expand \$k\$ paths at each decoding step.
>   * *MCTS* performs expansion and PRM-based scoring throughout.
>   * *Adaptive* uses serial sampling with adaptive stopping, which inhibits GPU parallelism, and it still requires reward models;
>     All three incur significantly higher computational cost compared to *ST-BoN*, which benefits from GPU-friendly parallel sampling and early truncation.
>
> * **Cost-performance Trade-off:** ST-BoN demonstrates a clear advantage:
>
>   * By progressively increasing $N$, ST-BoN can achieve the same performance level as the other three baselines while maintaining substantially lower computational cost
>   * Under the same cost, ST-BoN still performs better.
>
> **These results further reinforce the "cost" and "cost-performance trade-off" contribution of our ST-BoN.**
>
> ---
>
> > Q2: Generative reward models [3,4] which reduce the flop usage are not compared against.
>
> The comparison with GRM can be naturally included under the “Full-BoN w/ RM” baseline, treating it as an alternative reward model. Specifically, we selected **Gemma-7B-GRM** as proposed in [3]. Results are shown in the summary table in W1:
>
> * **Cost:** GRM outperforms RM in terms of accuracy, but its outputs are in natural language, which are significantly longer --- resulting in higher computational cost compared to RM.
>
> * **Cost-performance Trade-off:** ST-BoN continues to demonstrate a strong advantage.
>   * For example, on GPQA, when reaching an accuracy of ~36.5%, ST-BoN incurs only **3.7/14.4 = 25.7%** of the computational cost of Full-BoN w/ GRM.
>   * Under the same cost budget, ST-BoN clearly outperforms Full-BoN w/ GRM.
>
> **These results further reinforce the "cost" and "cost-performance trade-off" contribution of our ST-BoN.**
>
> ---
>
> > Q3: Compare to the rule-based (oracle) reward model to evaluate the bias of the PRM.
>
> Thanks for bringing up the idea of rule-based reward models. Since no specific reference was provided, we investigated some work and found that [7] uses predefined features and trains probes to model rewards in the safety domain, and [8] introduces an agent system that filters verifiable signals like instruction-following, fluency, and factuality. These work may align with rule-based reward models you mentioned.
>
> However, such approaches have not been extended to the reasoning domain, where correctness is essentially the only feature that matters. If one fits a reward function based on correctness, it may essentially reduce the method back to the PRM paradigm, thereby blurring the distinction between the two. Moreover, our task is not based on a formal language, so automatic evaluation tools similar to code execution engines cannot be applied.
>
> As an alternative, we used a more approximate oracle: **LLM-as-a-Judge**, since larger LLMs are typically more capable. We chose the latest ChatGPT-4o to evaluate the correctness of all sampled outputs, selecting the most frequently judged correct answer as the final output. The cost of using the Judge LLM includes both inference latency and API token usage.
>
> * *GPQA in Qwen2.5-7B-Instruct (Accuracy / Cost):*
>
> ||$N=5$|$N=10$|$N=20$|$N=30$|$N=60$|$N=80$|
> |-|-|-|-|-|-|-|
> |Full-BoN w/ RM|33.8 / 4.9x|34.3 / 6.6x|35.3 / 10.0x|35.7 / 13.1x|——|——|
> |Full-BoN w/ Oracle|34.1 / 7.6x + 4.2k tokens|35.1 / 13.9x + 8.8k tokens|**36.2 / 18.9x + 20.1k tokens**|36.6 / 30.2x + 37.1k tokens|——|——|
> |ST-BoN|32.3 / 1.3x|33.3 / 1.4x|33.5 / 1.7x|34.9 / 2.0x|36.1 / 3.1x|**36.4 / 3.7x**|
>
> * **Cost:** Introducing an Oracle as the reward model does lead to improved performance, but it also significantly increases computational cost --- including inference latency and API usage --- which runs counter to our motivation of "optimizing the best-of-N paradigm".
>
> * **Cost-performance Trade-off:** When achieving an accuracy of around 36.2, and excluding API overhead, the computational cost of ST-BoN is only **3.7 / 20.1 = 18.4%** that of Full-BoN w/ Oracle.
>
> **These results further reinforce the "cost" and "cost-performance trade-off" contribution of our ST-BoN.**
>
> ---
>
> > Q4: Evaluate how an RM trained on the summarization task would compare.
>
> First, we included the summarization task mainly to **show ST-BoN’s generalization ability**. Training reward models across domains is challenging due to the lack of quality preference data—for example, there’s no standard reward model for summarization. **ST-BoN avoids relying on such priors, making it more scalable and adaptable across tasks.**
>
> Despite this, we are happy to add a summarization-specific RM for clearer comparisons. Since no reward model and preference data exist for this task, we fine-tuned one based on ArmoRM-Llama-3-8B[6]. Using ChatGPT-4o, we annotated 6,000 samples from the XSum training set with 5-point ratings to train this summarization-specific RM.
>
> We then conducted a human evaluation comparing three settings: Full-BoN w/ RM (our original baseline), Full-BoN w/ RM-Sum (the fine-tuned summarization reward model), and ST-BoN. The results are shown below:
>
> * *CNNDM in Qwen2.5-7B-Instruct (Accuracy / Cost):*
>
> ||$N=5$|$N=10$|$N=15$|$N=20$|$N=30$|$N=60$|
> |-|-|-|-|-|-|-|
> |Full-BoN w/ RM|**4.30 / 3.2x**|4.30 / 5.0x|4.30 / 6.8x|4.34 / 8.6x|4.40 / 12.0x|——|
> |Full-BoN w/ RM-Sum|**4.34 / 3.1x**|4.34 / 4.7x|4.37 / 6.9x|4.40 / 8.5x|**4.43 / 11.8x**|——|
> |ST-BoN|4.33 / 1.6x|4.35 / 2.0x|4.38 / 2.3x|4.38 / 2.6x|4.42 / 3.3x|**4.43 / 4.3x**|
>
> The reward model fine-tuned on summarization data does yield improvements over its pre-trained counterpart. However, **ST-BoN continues to show a clear advantage in cost-performance trade-off**. To achieve a human rating of 4.43, ST-BoN requires only **4.3/11.8=36.8% of the cost** compared to Full-BoN w/ RM-Sum. Furthermore, at a comparable computational budget (~3×), ST-BoN **significantly outperforms** Full-BoN w/ RM-Sum in output quality.
>
> **These results further reinforce the "cost" and "cost-performance trade-off" contribution of our ST-BoN.**
>
> ---
>
> > Q5: Evaluation with Long-CoT models.
>
> ***<Please refer to Reviewer GGmq’s Q2 for a detailed response.>***
>
> **Overall, these results provide strong evidence for the scalability and efficiency of ST-BoN in the context of long-CoT reasoning models.**
>
> ---
>
> We hope that these additional results adequately address your concerns. We look forward to your positive feedback.
>
> [6] Interpretable Preferences via Multi-Objective Reward Modeling and Mixture-of-Experts
>
> [7] Rule Based Rewards for Language Model Safety
>
> [8] Agentic Reward Modeling: Integrating Human Preferences with Verifiable Correctness Signals for Reliable Reward Systems

---

### Official Review · Reviewer_QemV · 2025-07-02

**Clarity:** 3
**Significance:** 3
**Originality:** 3
**Rating:** 5
**Confidence:** 4

**Summary:**

The work introduces Self-Truncation Best-of-N, a novel decoding method for efficient Best-of-N sampling in LLM. Traditional BoN sampling has significant memory and latency costs due to full generation of multiple outputs and the use of reward models. To address these issues, the proposed ST-BoN technique identifies promising samples early during decoding by leveraging internal consistency metrics derived from latent space representations. This allows the method to truncate suboptimal sampling paths early, significantly reducing computational overhead without compromising performance. Extensive experiments demonstrate that ST-BoN achieves a superior cost-performance balance, offering substantial efficiency improvements while maintaining or improving accuracy across diverse tasks and domains.

**Questions:**

See in weakness.

**Ethical Concerns:**

["NO or VERY MINOR ethics concerns only"]

**Final Justification:**

Thank you for addressing some of my concerns. I raised my score to 5.

**Limitations:**

See in weakness.

**Quality:**

3

**Strengths And Weaknesses:**

Strengths:
1. The proposed early truncation based on internal state consistency is novel, distinguishing it from existing BoN optimization methods that mostly depend on external reward models or complete generation of samples.
2. The theoretical results (early consistency foreshadowing final consistency) are well demonstrated and support the proposed method. The extensive empirical evaluation across benchmarks is robust and thorough.

Weakness:
1. The experimental results primarily focus on relatively smaller models (7B parameter scale). Additional demonstrations of the effectiveness of ST-BoN on larger-scale models would further strengthen the paper’s claims regarding generalizability and scalability.
2. While generally clear, certain technical details such as specific implementation considerations or hyperparameter selection (e.g., setting τ) could benefit from additional intuitive explanations to help readers replicate or adapt the method more easily.
3. Although novel, the core idea—leveraging internal model states for decision-making during sampling—builds incrementally upon previous work using consistency strategies and latent embeddings. I know some works about early termination or convergence analysis in search algorithms, which is not included in related works or baselines. (This work can be considered as one-step search algorithm)

---

> ### Author Rebuttal · Authors · 2025-07-27
>
> Thank you for your recognition of our work, especially the novelty of the proposed method and the strength of both the theoretical analysis and empirical evaluation. Regarding the weaknesses you raised (W1 - W3), we address them one by one as follows.
>
> ---
>
> > W1: The experimental results primarily focus on relatively smaller models (7B parameter scale)
>
> Thank you for pointing this out. We kindly clarify that **our paper already includes experiments using models with 70B+ parameters (Qwen2.5-72B-Instruct)**, as noted by *Reviewer ciSh in Strength-4 "A study of robustness to increasing model size has been performed and reported on in Appendix D.1."*
>
> Due to space constraints in the main text, these results are presented in **Figure 10 of Appendix D.1**. The experimental results demonstrate that our method remains effective even on larger-scale models.
>
> ---
>
> > W2: Technical details such as specific implementation considerations or hyperparameter selection (e.g., setting $\tau$)
>
> Thank you for pointing this out. $\tau$ is the only hyperparameter in ST-BoN, and **we have already provided its intuitive explanation (Section 3.2) and selection rationale (Section 6.2) in the paper**. Here, we present the key information:
>
> * Intuitive Explanations I: Why set buffer window $\tau$ ?
>
> The intuitive explanation here can be referred to in the original text from **lines 140-142**: *Performing self-estimation at a single moment can introduce randomness, since pairwise sequence differences only begin to emerge at time $c$, and these differences may not be substantial.* This is verified in Section 6.1.
>
> * Intuitive Explanations II: Why define $\tau$ as $mc$ ?
>
> The intuitive explanation here can be referred to in the original text from **lines 143–147**: "*We set $\tau \propto c$, *i.e.*, $\tau = mc$, where $m$ is a proportional constant. The intuition is that a smaller $c$ indicates an early divergence between samplings, which reflects greater randomness, so a smaller window may suffice to capture later differences. In contrast, a larger $c$ suggests lower randomness, which requires a larger window to capture sufficient later difference.*"
>
> * Hyperparameter Selection: Why select $\tau$ as $c$ ?
>
> The hyperparameter selection rationle here is demonstrated by the ablation study in Section 6.2. **Figure 7** presents an ablation study on the cost–performance trade-off across four datasets by varying the value of $\tau$. As concluded in **lines 313–315**: *Figure 7 shows that when $\tau < c$, the performance gain relative to cost increases faster. However, beyond $c$, this gain slows significantly, making $c$ the optimal choice for balancing performance and cost in our experiments."*
>
>
> ---
>
> > W3: Although novel, the core idea builds incrementally upon previous work using consistency strategies and latent embeddings; I know some works about early termination or convergence analysis in search algorithms, which is not included in related works or baselines
>
> > *(1) the core idea builds incrementally upon previous work*
>
> Thank you for recognizing the novelty of our method. Regarding the incremental construction of the core idea, we believe that **the core idea of ST-BoN lies in *"the design of a complete self-truncation framework* that is free of full generation and reward models"**, but not only the consistency metric itself:
>
> * Prior to Section 3.2, our focus was entirely on how to eliminate the use of full generation and the reward model in BoN, without mentioning any details about specific consistency metrics such as the “chain-of-embedding” (includes Figure 1, which only illustrates the framework). **All of our motivation and theoretical modeling are aimed at proposing this framework, rather than any particular metric.** This constitutes the core of ST-BoN and serves as a fundamental premise.
>
>
> * Under this premise, the early consistency algorithm introduced in Section 3.2 **should be viewed more as an empirical contribution. It makes the theoretically sound self-truncating framework proposed in Section 3.1 and Figure 1 practically implementable**. From this perspective, whether the metric design is entirely novel or inspired by prior work is not essential; refining an already validated metric could even lend greater credibility to the overall approach.
>
> From another perspective, **the core contribution of ST-BoN --- efficiency improvement --- does not stem from any particular consistency metric, but rather from the self-truncating framework itself**. As noted by *Reviewer GGmq: “The overhead from BoN is a critical issue, and the proposed method does not rely on any additional model or training to successfully reduce the overhead, which is significant and insightful and can motivate more work exploring consistency during sampling.”* --- it is this framework design that holds the most promise for inspiring further work. **Therefore, the overall framework of ST-BoN and the consistency metric are equally important and mutually reinforcing in their contributions.**
>
> > *(2)  I know some works about early termination or convergence analysis in search algorithms, which is not included in related works or baselines. (This work can be considered as one-step search algorithm)*
>
> We thank you for pointing out related work on early termination or convergence analysis in search algorithms. For example, [1] has explored pruning strategies in beam search.
>
> However, we would like to clarify that **our goal is to optimize the *best-of-N* paradigm itself, which is an independent search paradigm (randomized sampling search)**. Other search paradigms, such as beam search and MCTS, are representative of heuristic-guided search. These paradigms differ substantially in both algorithmic structure and optimization dynamics, and we consider them **orthogonal research directions**.
>
> **In our current version, we have already provided an in-depth discussion of early termination techniques within the Best-of-N search framework.** That said, we fully acknowledge the value of cross-paradigm insights. We will carefully consider your suggestion and plan to include a broader discussion of early termination strategies in other search paradigms in the next revision.
>
> ---
>
>
> We hope these clarifications address your concerns and provide a clearer understanding of our arguments, contributions, and empirical conclusions. We sincerely appreciate your constructive comments and look forward to positive feedback.
>
> [1] Beam search pruning in speech recognition using a posterior probability-based confidence measure.

---

> > ### Comment · Reviewer_QemV · 2025-08-05
> >
> > Thank you for addressing some of my concerns. I raised my score to 5.

---

> > > ### Author Response · Authors · 2025-08-06
> > >
> > > We are glad to see that we have addressed your concerns, and we sincerely appreciate your decision to raise the score in support of our work.

---

### Official Review · Reviewer_ciSh · 2025-07-03

**Clarity:** 3
**Significance:** 3
**Originality:** 2
**Rating:** 5
**Confidence:** 3

**Summary:**

The authors propose a decoding method that eliminates the need for reward models by leveraging early sampling consistency to identify and truncate suboptimal paths. The method is claimed to significantly reduce dynamic GPU memory usage by over 80% compared to Full-Best-of-N by truncating suboptimal samples in early decoding, having the implication of reducing inference latency and changing the parameters of the cost-performance trade-off. The key part of the method is the design of an internal consistency measure that is used to choose the most promising sample in early decoding. The authors also claim that the method remains effective as model size increases.

**Questions:**

See weaknesses.

**Ethical Concerns:**

["NO or VERY MINOR ethics concerns only"]

**Final Justification:**

In the light of the author response and the promise to address my concerns in a revision of the paper I raise my score.

**Limitations:**

A short discussion of limitations is given in Appendix F. I did not spot any explicit discussion of the paper's societal impact, though given the nature of the work and the prevalence of reasoning models in popular use, I would expect at least some.

**Paper Formatting Concerns:**

None.

**Quality:**

3

**Strengths And Weaknesses:**

**Strengths**
1. Contemporary relevance of the paper's topic.
2. The work is well-intended to replace the additional dependency of reasoning systems on reward models.
3. The work is well-grounded with a theoretical justification for the proposed approach.
4. A study of robustness to increasing model size has been performed and reported on in Appendix D.1.
5. A related work section following the main claims of the text that clearly points to differences between the results of the paper and the related work.


**Weaknesses**
1. The experimentation of Section 5 does not conclusively demonstrate the universality of the internal consistency measure.
2. I did not spot any explicit discussion of the paper's limitations or societal impact in the main text, though given the nature of the work and the prevalence of reasoning models in popular use, I would expect at least some. The authors should answer the following question: how does the use of the generating trajectory sampling method they propose going to affect models trustworthiness, safety, and inherent biases? A proper discussion of limitations compiled and moved from Appendix F.

---

> ### Author Rebuttal · Authors · 2025-07-27
>
> Thanks for your recognition of our work, particularly the well-motivated method, sound theoretical justification, robust experimentation, and comprehensive coverage of related work. Below, we address the weaknesses (W1 - W2) you raised one by one:
>
> ---
>
> > W1: The experimentation of Section 5 does not conclusively demonstrate the universality of the internal consistency measure.
>
> Thank you for pointing this out. We agree that the experiments in Section 5 provide a comprehensive cost–performance evaluation but do not directly isolate the contribution of individual components, such as the role of internal consistency measure.
>
> **To address this, we conducted a fine-grained ablation study in Section 6.1 (Figures 6, 14, and 15) to assess the universality of the internal consistency measure.** Specifically, we replaced the "internal consistency measure" with two "output-based consistency measures" --- *semantic similarity* and *string overlap* --- and evaluated the *correlation between early self-estimation results and final correctness*. We conducted experiments across three models and found that **only our internal consistency measure consistently achieved correlations significantly above random, demonstrating its universality.**
>
> ---
>
>
> > W2: The paper's societal impact: how does the use of the generating trajectory sampling method they propose going to affect models trustworthiness, safety, and inherent biases?
>
> We appreciate this important discussion. Societal implications are indeed crucial for applied research, and we summarize our perspective below:
>
> * **Trustworthiness:** ST-BoN relies on internal consistency signals to guide decoding, moving the selection process into the model's latent space. Prior work [1] has shown a strong correlation between consistency and correctness. Building on this, our theoretical analysis (Section 3.1) and empirical findings (Section 6.1) further support the reliability of latent-space decision-making. Together, these two parts provide a solid foundation for the model trustworthiness under our method.
>
>
> * **Safety:** ST-BoN introduces no additional safety risks. The sampling follows the model’s native distribution without adversarial perturbation, and the consistency-based selection mechanism operates without external reward models, thus avoiding interference from out-of-distribution signals.
>
>
> * **Biases:** ST-BoN inherits the base model’s biases, as it does not perform explicit debiasing. However, by leveraging multiple parallel samplings, it can identify and truncate transient biases that arise sporadically during generation. These biased trajectories may be excluded as outliers through the consistency-based strategy, thereby potentially mitigating their impact.
>
>
> We will incorporate these points into an additional “Societal Impact” section in the revised version of the paper.
>
> ---
>
> We hope these clarifications address your concerns and provide a clearer understanding of our empirical conclusions. We sincerely appreciate your constructive comments and look forward to positive feedback.
>
>
> [1] Self-consistency improves chain of thought reasoning in language models.

---

### Decision · Program_Chairs · 2025-09-17

**Decision:**

Accept (spotlight)

**Comment:**

The paper studies improving the inference efficiency of Best-of-$N$ (BoN) sampling through early pruning. Previous methods for early pruning is mainly based on evaluation from a process reward model, which requires additional compute. The paper proposes a pruning method using self-consistency of the chain-of-embedding (CoE) features. The paper provides justifications on the proposed method by showing that early consistency could indicate consistency in full generation, and hence it could be used as a metric to select promising candidates. The paper empirically demonstrates the proposed method by comparing to full BoN baselines with and without reward models and demonstrate significant efficiency advantage with comparable performance. Reviewers all agreed that the paper provides a novel, principled approach for efficient BoN implementations. Concerns were raised about the coverage of empirical evaluation regarding different models, and baseline methods. The authors addressed these in the rebuttal and the reviewers are satisfied.

Overall, I believe this is a valuable contribution to the field. I believe the idea of using model’s internal states to perform self-truncating when generating multiple inference traces is interesting idea, which is worth highlighting at the conference. Hence I will would an spotlight acceptance.